JCB Journal of Cell Biology

# DNA topoisomerase 3 is required for efficient germ cell quality control

Maria Rosaria Dello Stritto[1], Bernd Bauer[1], Pierre Barraud[2], and Verena Jantsch[1]

**An important quality control mechanism eliminates meiocytes that have experienced recombination failure during meiosis. The culling of defective oocytes in *Caenorhabditis elegans* meiosis resembles late oocyte elimination in female mammals. Here we show that topoisomerase 3 depletion generates DNA lesions in both germline mitotic and meiotic compartments that are less capable of triggering p53 (*cep-1*)–dependent apoptosis, despite the activation of DNA damage and apoptosis signaling. Elimination of nonhomologous, alternative end joining and single strand annealing repair factors (CKU-70, CKU-80, POLQ-1, and XPF-1) can alleviate the apoptosis block. Remarkably, the ability of single mutants in the other members of the Bloom helicase-topoisomerase-RMI1 complex to elicit apoptosis is not compromised, and depletion of Bloom helicase in topoisomerase 3 mutants restores an effective apoptotic response. Therefore, uncontrolled Bloom helicase activity seems to direct DNA repair toward normally not used repair pathways, and this counteracts efficient apoptosis. This implicates an as-yet undescribed requirement for topoisomerase 3 in mounting an effective apoptotic response to ensure germ cell quality control.**

## Introduction

Quality control through oocyte elimination is a widely conserved mechanism that plays a pivotal role in preventing the formation of faulty gametes (Hunter, 2017). The DNA damage response and induction of apoptotic cell death are temporally and spatially regulated in the *Caenorhabditis elegans* germline. Germline meiocytes residing at different stages of the cell cycle respond differently to exogenous genotoxic insults or the presence of aberrant recombination intermediates. Only cells in late pachynema can undergo apoptosis because CEP-1 (the p53 orthologue) is translationally repressed until that stage (Schumacher et al., 2005). Thus, apoptosis seems to be triggered in well-defined spatial and temporal windows during gamete maturation. A *cep-1*–dependent checkpoint induces apoptosis in the presence of persistent DNA damage (Gartner et al., 2008). CEP-1 activates the transcription of EGL-1 (Schumacher et al., 2001), which in turn promotes the release of CED-4 (the worm orthologue of Apaf1), culminating in the activation of CED-3, the major caspase necessary for apoptosis (Conradt and Horvitz, 1998).

The morphological changes that occur in dying cells have been characterized at unprecedented resolution in worms (Huelgas Morales and Greenstein, 2018). For example, the export of mitochondria along microtubule cables into the gonad cytoplasmic core coincides with alteration of the P granule content around the nuclear membranes (Raiders et al., 2018). Nuclear-to-cytoplasmic translocation of the SIR-2.1 deacetylase is an early apoptotic event (Greiss et al., 2008). Even in the absence of persistent DNA damage, about half of the nuclei in pachynema undergo cell death under normal physiological conditions (Gumienny et al., 1999) to provide a supply of nutrients and organelles for the developing oocytes (Andux and Ellis, 2008; Raiders et al., 2018). Apoptosis in response to meiotic failure is comparable to late oocyte attrition that occurs in mammals (Gartner et al., 2008).

Two major biological cell division processes contribute to the generation of healthy gametes: (1) the mitotic proliferation of premeiotic cells, which then (2) undergo meiosis to reduce the genome content and shuffle the parental genomes. During meiosis, two rounds of cell division follow a single DNA replication step. Pairs of homologous chromosomes separate during meiosis I. In meiosis II, each chromosome splits into its two sister chromatids. During prophase I, several processes lead to the formation of a physical linkage between homologues, which aids faithful chromosome segregation: these include regulated double-strand break (DSB) induction, homologue juxtaposition, stabilization of aligned homologues by the synaptonemal complex, DNA repair via homologous recombination (HR) to produce crossovers (COs), and the installation of cohesion (Gerton

[1]Department of Chromosome Biology, Max Perutz Laboratories, University of Vienna, Vienna Biocenter, Vienna, Austria; [2]Expression Génétique Microbienne, UMR 8261, Centre Nationale de la Recherche Scientific, Université de Paris, Institut de Biologie Physico-Chimique, Paris, France.

Correspondence to Verena Jantsch: verena.jantsch@univie.ac.at.



and Hawley, 2005; Zickler and Kleckner, 1999). These processes can be easily followed in the *C. elegans* hermaphrodite syncytial gonad over a spatiotemporal course, in which cells from the mitotic compartment enter meiosis and progress through the stages of leptonema/zygonema to pachynema and diplonema to form cellularized oocytes in diakinesis.

The major error-free pathway, HR, is the predominant DSB repair pathway used during meiosis. Following the induction of DSBs, DNA resection generates single-stranded overhangs at the DNA break that are stabilized by replication protein A (RPA). The subsequent replacement of RPA by the RAD-51 recombinase generates RAD-51–covered single-stranded DNA (ssDNA) fibers, which invade one chromatid of the parental homologue (forming D-loop structures) and use it as a repair template during HR-mediated repair. After DNA synthesis, a DNA double Holliday junction (dHJ) molecule is generated. The dJH is resolved as either CO or non-CO depending on the directionality of DNA cleavage by the resolvases (nucleases that cleave dHJs), and both of these are outcomes of recombination (Hunter, 2015). The STR/BTR complex, composed of the Sgs1/Bloom helicase, topoisomerase 3, and scaffolding proteins (RMI1 and, in some organisms, RMI2), greatly aids the recombination process at multiple steps. Topoisomerase 3 has two enzyme activities that operate preferentially on hemicatenane DNA structures: cleavage of one DNA strand and passage of the other DNA strand (Haber, 2015). Prominent activities in meiosis are the displacement of invaded D-loops (which prevents multi-chromatid joint molecules) and decatenation of joint DNA molecules (which would otherwise impede chromosome segregation; Kaur et al., 2015; Kaur et al., 2019; Tang et al., 2015). The STR/BTR complex can dismantle dHJ substrates in vitro, reflecting its essential activity of reinforcing non-CO outcomes during mitotic recombination (Bizard and Hickson, 2014). In addition, the complex has an unrelated and poorly understood role in supporting COs while they are being formed (Jagut et al., 2016; Schvarzstein et al., 2014). Topoisomerase 3 also has a role in the repair of replication lesions (Cejka et al., 2010) and the final separation/resolution of replicated circular mitochondrial genomes (Nicholls et al., 2018).

When HR is compromised, error-prone DNA repair pathways, which are usually suppressed in the germline (such as nonhomologous end joining [NHEJ], alternative NHEJ [Alt-NHEJ], or single-strand annealing [SSA]) can be employed as salvage pathways to repair DSBs (e.g., see Macaisne et al., 2018; Martin et al., 2005). In NHEJ, the broken ends are first captured by the heterodimer CKU-70/80 and then ligated by DNA ligase IV (LIG-4; Chapman et al., 2012). Despite preferential use of the HR pathway, CKU-70/80 has been shown to load onto meiotic breaks, with active replacement of the dimer enabling resection (Girard et al., 2018; Lemmens et al., 2013). If resection is limited, short ssDNA close to the DNA break can generate a substrate for polymerase theta (POLQ-1) to engage in the Alt-NHEJ pathway (Boulton and Jackson, 1996; Schimmel et al., 2017). DNA lesions with long resected ssDNA tracks can employ SSA, which involves the endonuclease XPF-1 (e.g., Lans et al., 2013; Macaisne et al., 2018).

The UFD-2 ubiquitin ligase coordinates the decision between persistent DNA damage signaling and apoptosis by promoting the dissociation of RAD-51 from DNA repair sites to expose RPA-1–covered ssDNA lesions; persistently high RAD-51 levels lead to suppression of apoptosis (Ackermann et al., 2016).

*C. elegans* topoisomerase 3 (*top-3*) mutants display replication defects and meiotic accumulation of undefined DNA masses that cannot segregate to produce viable eggs (this study and Kim et al., 2002; Wicky et al., 2004). Here we show that *top-3* germline nuclei accumulate an impressive amount of RAD-51 and RPA-1. Surprisingly, despite the initial up-regulation of DNA damage signaling, followed by EGL-1 activation and the induction of apoptosis-associated events, the apoptotic levels in *top-3* mutants did not exceed the levels seen in WT gonads. Further analysis reveals that the efficient execution of DNA damage–induced apoptosis can be bypassed by either premeiotic depletion of RAD-51 or by depletion of HIM-6, CKU-70/CKU-80, POLQ-1, or XPF-1 in *top-3*. We propose that in *top-3* mutants, uncontrolled activity of the helicase HIM-6 produces aberrant recombination intermediates that activate the NHEJ/Alt-NHEJ/SSA pathways. We speculate that once those aberrant recombination intermediates engage in one of these alternative DNA repair pathways, the DNA damage signaling is not sufficiently sustained to reinforce the efficient culling of unhealthy oocytes. Thus, this study has identified an important role for topoisomerase 3 in germ cell quality control.

## Results

### *C. elegans* DNA topoisomerase 3 mutants accumulate DNA lesions

Phenotypic differences in meiosis were observed between mutants of the various STR/BTR complex members, resulting in different diakinesis chromosome configurations (Agostinho et al., 2013; Jagut et al., 2016; Kim et al., 2000; Schvarzstein et al., 2014; Wicky et al., 2004). To investigate the role of *C. elegans top-3* in meiosis, we generated the gene disruption allele *top-3 (jf101)* (Fig. 1 A).

The *top-3* mutants displayed 100% embryonic lethality and a prominent reduction in brood size. Adult worms laid an average of only 27.3 ± 8.1 (mean ± SD) embryos compared with the average brood size of 237.6 ± 4 for the WT (Fig. 1 B). Similar to *top-3* RNAi worms (Kim et al., 2000), the number of nuclei per gonad was reduced to 141.4 ± 12.5 (compared with 322.8 ± 23.7 in the WT; Fig. 1 C and Fig. S1 A), and the mitotic gonad compartment was prolonged (Fig. S1 A).

To examine the completion of meiotic prophase I events, we inspected and quantified the number of bivalents at diakinesis. At this stage, six DAPI-stained bodies (corresponding to homologous chromosome linked by chiasmata) are visible in the WT. In contrast, equivalent cells in the *top-3* mutant contain an average of 5.5 morphologically aberrant DAPI-stained bodies (Fig. 1 D). We next performed FISH of diakinesis chromosomes using markers for two different chromosomes and found that the abnormalities present at this stage were caused by the aggregation of nonallelic DNA filaments (Fig. S1 B). This finding is consistent with the presence of aberrant connections between chromosomes. The *top-3* diakinesis phenotype depends on meiotic DSBs generated by the SPO-11 topoisomerase. The *top-3;*

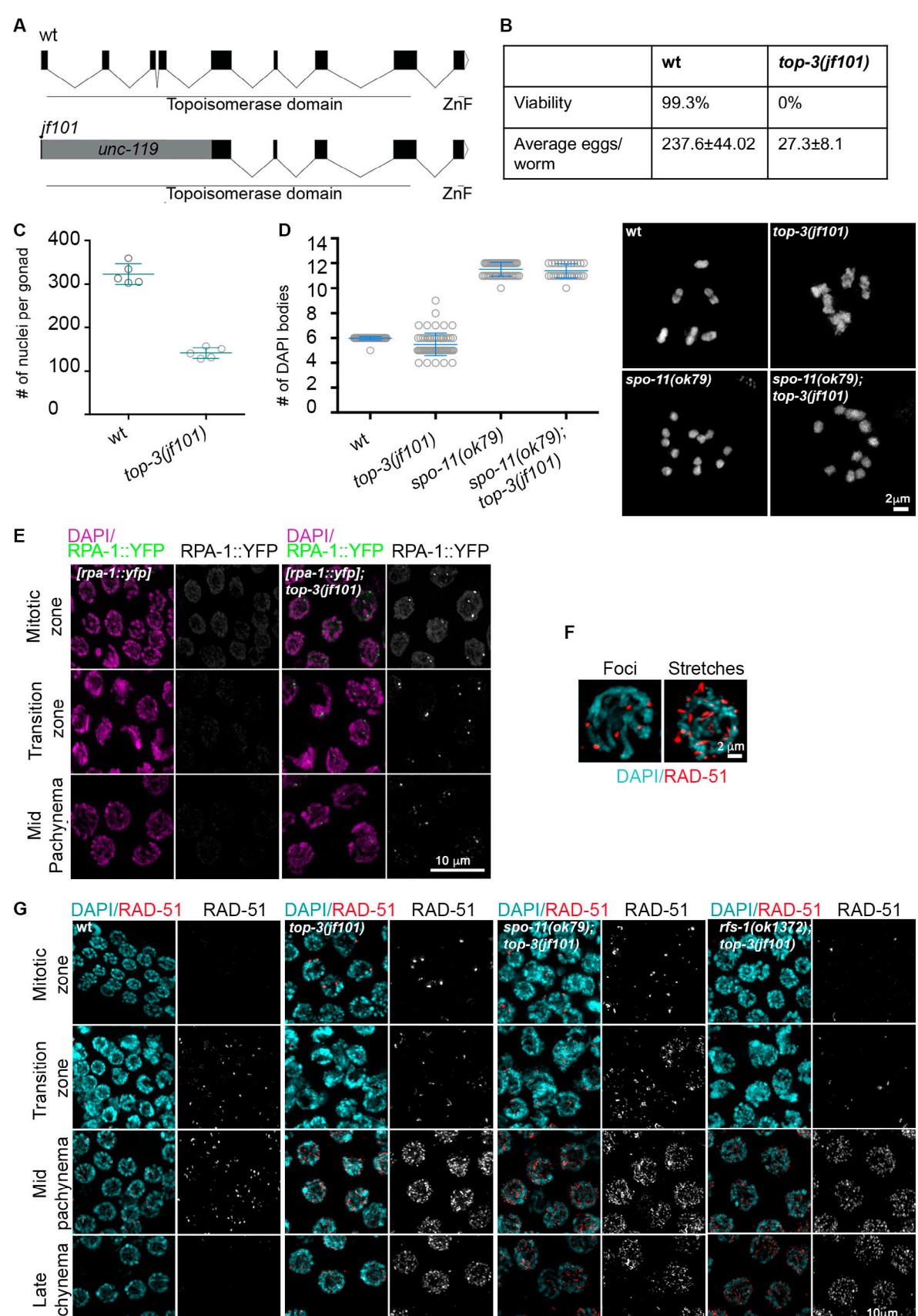

Figure 1. **Germline phenotype of the *top-3* mutant. (A)** Schematic representation of the *top-3* gene, indicating the conserved topoisomerase domain, the RGF zinc finger (ZnF) motif, and introns and exons. In the *jf101* allele, the ORF is disrupted by insertion of the *unc-119* locus directly after the start codon,

resulting in the deletion of 171 aa in the N terminus. **(B)** Hatch rates and brood sizes of the *top-3(jf101)* mutants (*n* = 15 worms) and wt (*n* = 16 worms). **(C)** Quantification of the total number of germline nuclei (from the mitotic region to late pachynema). Scatter plots indicate the mean ± SD number. Five gonads were quantified for each genotype. wt, 322.8 ± 23.7; and *top-3(jf101)*, 141.4 ± 12.5. **(D)** Quantification of DAPI bodies in the −1 diakinesis oocyte: wt, 6 ± 0.06, *n* = 57; *top-3(jf101)*, 5.4 ± 0.8, *n* = 65; *spo-11(ok79)*, 11.5 ± 0.5, *n* = 54; and *top-3(jf101); spo-11(ok79)*, 11.4 ± 0.6, *n* = 33. *n* is the number of oocytes. The scatter plot indicates the mean ± SD. Inserts show representative images of DAPI-stained diakinesis nuclei of the indicated genotypes. See Fig. S1 A for an additional phenotypic analysis. **(E)** Representative images of [RPA-1::YFP] localization at different stages of meiotic prophase I in the indicated genotypes. Gonads were stained with DAPI (magenta) and endogenous YFP (green). **(F)** Higher magnification images showing RAD-51 stretches in *top-3* mutants (right) and RAD-51 foci in the wt (left). **(G)** Representative images of RAD-51 foci localization at different stages of meiotic prophase I for the indicated genotypes. Gonads were stained with DAPI (cyan) and an anti-RAD-51 antibody (red). wt, wild-type.

*spo-11* mutant displayed 11.4 ± 0.6 (mean ± SD) DAPI-stained bodies, corresponding to 12 achiasmatic chromosomes (*spo-11*: 11.5 ± 0.5 DAPI-stained bodies; Fig. 1 D). This result suggests that the joint DNA molecules accumulate only after the induction of meiotic DSBs.

The phenotype of the *top-3* null mutant differs from the phenotypes of null mutants of the other complex members (*rmh-1* [worm RMI1]; *him-6* [worm bloom helicase]) in that embryonic death is more severely affected in *top-3* (viability: *rmh-1*, 26.5 ± 7.6%; *him-6*, 43 ± 1.7%; Agostinho et al., 2013; Jagut et al., 2016; Schvarzstein et al., 2014; Wicky et al., 2004). Furthermore, the formation of joint aberrant DNA structures at diakinesis could still be observed (although the number of DAPI bodies varied), even when the major or alternative repair pathways known to be employed in the worm gonad were impaired, as follows: HR, by knocking down the sumo/ubiquitin ligase *zhp-3* and the endonuclease *mus-81*; NHEJ, by knocking down the DNA-binding protein *cku-80*; Alt-NHEJ, by knocking down the polymerase *polq-1*; sister chromatid exchange, by knocking down the ubiquitin ligase *brc-1*; and synthesis-dependent strand annealing, by knocking down the endonuclease *xpf-1* (Fig. S1, C and D; Macaisne et al., 2018). Persistent catenane structures are obvious in the *top-3; zhp-3* double mutants, where 12 univalents would be expected (Fig. S1 D).

To determine whether the *top-3* phenotype is caused by problems during early DSB repair events, we assessed the kinetics of RPA-1 and RAD-51 foci dynamics during prophase I (Alpi et al., 2003; Colaiácovo et al., 2003; Stergiou et al., 2011). We detected the accumulation of the ssDNA-coating protein RPA-1 and/or the recombinase RAD-51, indicating that ssDNA tails were not being correctly processed at an early stage of recombination. In *top-3*, bright and distinct RPA-1 foci were localized throughout the gonad, a feature that is never seen in the WT when using this particular RPA-1 transgene with the given level of resolution (Fig. 1 E). Likewise, RAD-51 foci accumulation started in the mitotic zone and continued throughout the gonad, where numerous foci persisted even up to diplonema (also see Wicky et al., 2004). Moreover, RAD-51 foci were much more abundant and coalesced into stretches, which is never observed in the WT (Fig. 1, F and G). We found that the formation of RAD-51–coated DNA recombination intermediates was not dependent on meiotic DSBs, in contrast to the diakinesis chromosome phenotype (Fig. 1 G and Fig. S2 A). RAD-51 foci were not detected in the *spo-11* mutant; however, RAD-51 accumulation in *spo-11; top-3* was similar to the *top-3* single mutant (Fig. 1 G and Fig. S2 A), suggesting that most of the accumulated RAD-51 foci derived from the premeiotic stage. However, these lesions found in *top-3* cannot serve as substrates for meiotic CO formation.

The *C. elegans* RAD-51 paralog, RFS-1, has been shown to promote HR at DNA lesions during replication. RFS-1 recruits RAD-51 to replication forks that have stalled by the action of DNA cross-linking agents and other replication blocking lesions, but is dispensable for the repair of meiotic DSBs (Ward et al., 2007). Depletion of RFS-1 in the *top-3* mutant prevented RAD-51 accumulation in the premeiotic region (Fig. 1 G and Fig. S2 A). Nevertheless, RAD-51 accumulated in meiocytes in the *rfs-1 top-3 spo-11^{RNAi}* mutant (Fig. S2 A), indicating that the premeiotic aberrant recombination intermediates are carried into meiocytes.

Taken together with the finding that DNA lesions are observed in the *top-3; spo-11* double mutant, this result confirms that the absence of TOP-3 leads to abundant premeiotic unprocessed intermediates.

### *top-3* mutants activate DNA damage signaling but not efficient germline apoptosis

Exposure of *C. elegans* germlines to DNA damaging ionizing irradiation (IR; causing DSBs) triggers the activation of cell cycle checkpoints. In the gonad, DSBs induce cell cycle arrest in the proliferative mitotic zone and an increased number of nuclei entering apoptosis in late pachynema (Gartner et al., 2008; Gartner et al., 2000).

Cell cycle arrest in the mitotic gonad tip is caused by the inactivation of cyclin-dependent kinase (CDK-1), through phosphorylation by the CHK-1 (and, to a lesser extent, CHK-2) kinase. CDK-1 phosphorylation is downstream of ATR/ATM kinase (*C. elegans* ATL-1/ATM-1) activation by DNA damage (Fig. 2 A; Garcia-Muse and Boulton, 2005; Kalogeropoulos et al., 2004; Lee et al., 2010; O'Connell et al., 2000). In the *top-3* mutant, G2-arrested nuclei were seen in the gonad mitotic tip. These were positive for phospho–CDK-1 (Fig. 2 B), a feature only seen in the WT after irradiation. This result shows that DNA damage signaling is activated in the *top-3* mutant.

Physiological apoptosis eliminates around 50% of germ cells; however, persistent DNA damage increases apoptosis levels by directing nuclei toward apoptosis in late pachynema (Gartner et al., 2008). We quantified the number of apoptotic nuclei per gonad in live animals in two different ways: (1) using a CED-1::GFP transgene, which marks the engulfing cells that surround the apoptotic nucleus (Craig et al., 2012); and (2) incubation with SYTO-12, a dye that readily cross-reacts with nucleic acids in nuclei undergoing apoptosis. In contrast to its activation of cell cycle arrest in *top-3*, a high basal level of DNA damage did not increase apoptosis above the physiological level (or, in the

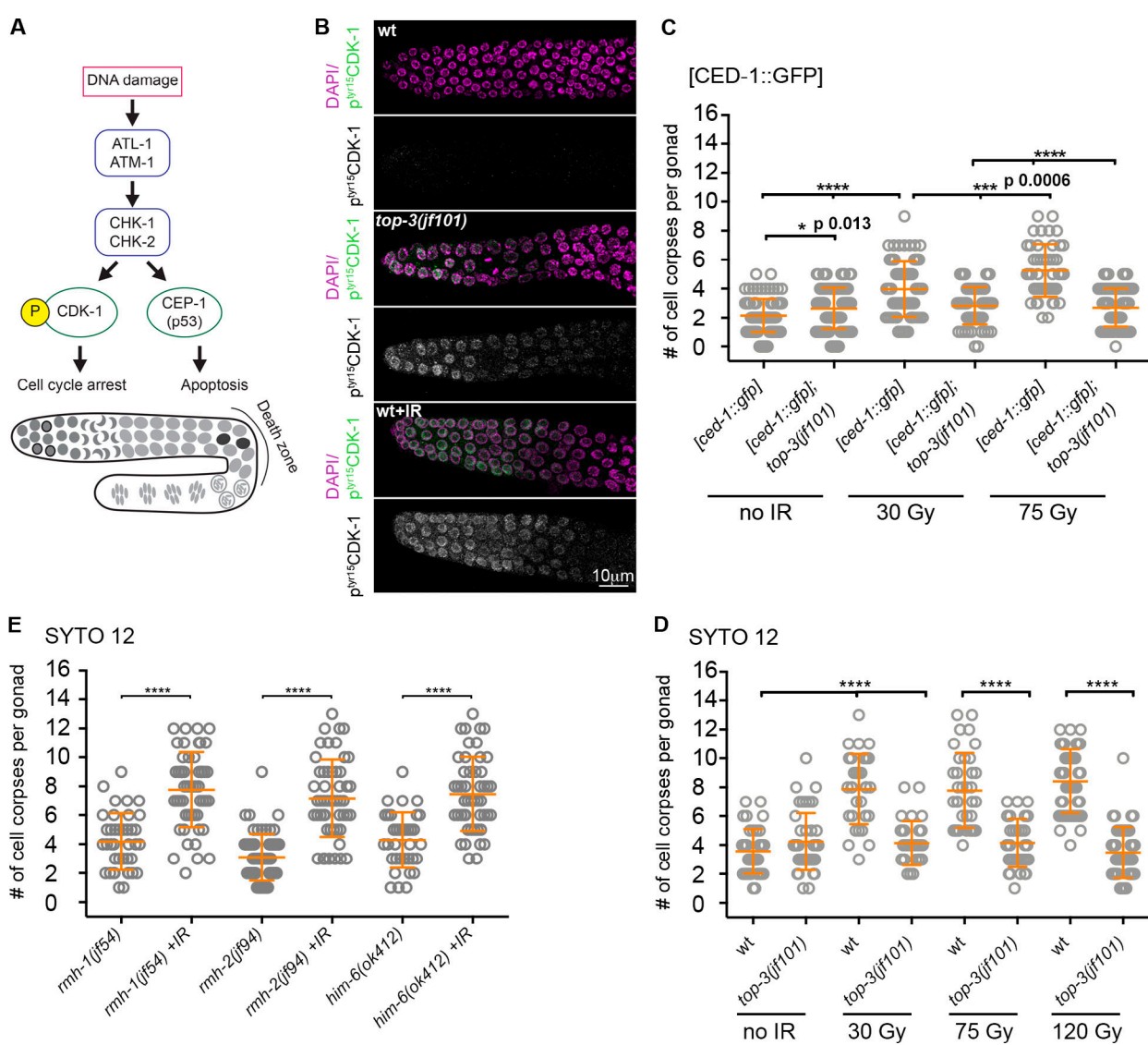

Figure 2. **The *top-3* mutant activates DNA damage signaling but fails to induce apoptosis. (A)** Schematic representation of DNA damage checkpoint activation in *C. elegans* meiosis. After the activation of the kinases triggered by DNA damage, CDK-1 phosphorylation leads to cell cycle arrest in the mitotic zone and activation of CEP-1 (p53) induces apoptosis in the pachynema. **(B)** p$^{tyr15}$CDK-1 localization in the mitotic zone in the wt, wt+ 120 Gy IR and in the undamaged *top-3* mutant. Gonads were stained with DAPI (magenta) and anti-pCDK-1 antibody (green). **(C and D)** Quantification of apoptotic nuclei using the markers [CED-1::GFP] (C) and SYTO-12 (D). Worms were exposed to 0, 30, 75, or 120 Gy IR, and apoptotic nuclei were scored after 20 h. Scatter plots indicate the mean ± SD. *[ced-1::gfp]* (no IR, 2.1 ± 1.1, *n* = 94 worms; 30 Gy, 4 ± 1.9, *n* = 68; 75 Gy, 5.3 ± 1.8, *n* = 46); *[ced-1::gfp]; top-3(jf101)* (no IR, 2.6 ± 1.4, *n* = 94; 30 Gy, 2.8 ± 1.3, *n* = 64; 75 Gy, 2.7 ± 1.3, *n* = 76); wt (no IR, 3.6 ± 1.5, *n* = 39; 30 Gy, 7.9 ± 2.4, *n* = 31; 75 Gy, 7.8 ± 2.6, *n* = 35; 120 Gy, 8.4 ± 2.2, *n* = 50); and *top-3(jf101)* (no IR, 4.2 ± 2, *n* = 43; 30 Gy, 4.1 ± 1.5, *n* = 31; 75 Gy, 4.1 ± 1.7, *n* = 35; 120 Gy, 3.5 ± 1.8, *n* = 46). Non-significant differences are not shown. ****, P < 0.0001, P values were calculated using the Mann–Whitney test.; **(E)** Apoptosis quantification using SYTO-12 in the indicated genotypes without irradiation and after exposure to 120 Gy IR. *n* = number of gonads scored: *rmh-1(jf54)*, 4.2 ± 1.9, *n* = 39; *rmh-1(jf54)*+IR, 7.8 ± 2.6, *n* = 51; *him-6(ok412)*, 4.3 ± 1.9, *n* = 35; *him-6(ok412)*+IR, 7.5 ± 2.6, *n* = 49; *rmh-2(jf94)*, 3.1 ± 1.6, *n* = 61; and *rmh-2(jf94)*+IR, 7.2 ± 2.7, *n* = 52. ****, P < 0.0001 calculated using the Mann–Whitney test. wt, wild-type.

case of CED-1::GFP quantification, only slightly increased the number of apoptotic cells; Fig. 2, C and D). As a comparison, persistent DNA damage increased the level of apoptosis by approximately threefold in the SCF ubiquitin ligase F box protein mutant *prom-1* (Jantsch et al., 2007; Fig. S2 B). Normalization of the number of apoptotic nuclei to the total amount of nuclei present in the last 10 cell rows of the gonad (the region competent for apoptosis execution) indicated that apoptosis is only partially induced by endogenous and exogenous DNA damage in *top-3* compared with irradiated WT worms or with *prom-1* mutants (Fig. S2 C).

To better understand whether the low levels of apoptosis in *top-3* are a consequence of reduced numbers of germline nuclei, we quantified the apoptotic cells in the Notch signaling mutant *glp-1(bn18)* (Fox and Schedl, 2015). Abrogation of the GLP-1 signal eliminates the pool of proliferative germ cells, resulting in a gonad with fewer nuclei (71.7 ± 8.8 [mean ± SD]) compared with the WT (333.7 ± 7.8; Fig. S2 E). In *glp-1*, endogenous lesions and DNA damage induced by irradiation both triggered the same apoptotic responses seen in WT worms (Fig. S2 D). We conclude that a low number of germ

cells does not necessarily mean a decreased apoptotic response.

Challenging with different IR doses did not increase the number of apoptotic nuclei in *top-3* mutants, but had a strong impact in control animals (Fig. 2, C and D). Whereas apoptosis levels increased above physiological levels in the WT after low-dose IR (30 Gy), higher doses did not change apoptosis levels in *top-3*, highlighting a defect in apoptosis. This contrasts with the apoptotic responses observed in mutants of other members of the STR/BTR complex (*rmh-1, rmh-2,* and *him-6*), which are competent to up-regulate apoptosis after irradiation (Fig. 2 E). This finding suggests that *top-3* has a specific defect in triggering efficient DNA damage–induced apoptosis.

We conclude that the presence of abundant DNA damage in *top-3* mutants cannot effectively trigger apoptosis. Moreover, in contrast to null mutants of other members of the STR/BTR complex, even high doses of irradiation cannot augment apoptosis in the *top-3* mutant.

### The apoptosis machinery is active in *top-3* mutants

As *top-3* mutants are competent to activate cell cycle arrest during S-phase in the proliferative zone but cannot effectively trigger germline apoptosis in response to DNA damage, we wanted to examine how and to what extent apoptosis could be initiated.

DNA damage–induced apoptosis depends on CEP-1 (the p53 orthologue) and its transcriptional targets, such as EGL-1 (Conradt and Horvitz, 1998; Derry et al., 2001; Hofmann et al., 2002; Schumacher et al., 2001). After CEP-1–mediated transactivation of *egl-1*, CED-4 becomes capable of activating the CED-3 caspase (Gartner et al., 2008; see the scheme in Fig. 3 A). CEP-1 is expressed in the proliferative region, transcriptionally repressed upon meiotic entry and turned on again in the "death zone" (late pachynema/diplonema; Hofmann et al., 2002; Schumacher et al., 2005). Immunostaining revealed that CEP-1 localization is similar in the *top-3* mutant and the WT (Fig. 3 B). Next, we examined whether the expressed CEP-1 was competent to activate *egl-1* transcription (Craig et al., 2012). For this, we analyzed *egl-1* expression by real time quantitative PCR (RT-qPCR) in WT worms before and after IR treatment and in undamaged *top-3* worms. We observed that *egl-1* transcription in *top-3* was constitutively up-regulated to levels comparable to those of irradiated WT worms (120 Gy; Fig. 3 C). These results suggest that CEP-1–mediated signaling is efficient in *top-3*.

In parallel, but independently of *cep-1*, the SIR-2.1 deacetylase is removed from nuclei that undergo apoptosis (Greiss et al., 2008). SIR-2.1 immunostaining confirmed the presence of SIR-2.1–free nuclei in the *top-3* germline (Fig. 3 D). Another event in cells undergoing apoptosis is the removal of PGL-1 from P granules (Raiders et al., 2018). PGL-1 immunostaining confirmed that PGL-1 is efficiently removed in *top-3* gonads (Fig. 3 E). The quantification of PGL-1–free nuclei and normalization to the total number of nuclei in the death zone revealed that PGL-1 removal in *top-3* gonads takes place at levels comparable to those of WT worms exposed to genotoxic stress (Fig. 3 E). Therefore, we conclude that the apoptotic machinery is activated in *top-3*.

### The apoptosis block in *top-3* is due to aberrant mitotic and meiotic recombination intermediates

In a mutant with persistent recombination intermediates, apoptosis execution was previously suggested to depend on RAD-51 removal from the recombination intermediates by a process involving the Ubiquitin Fusion Degradation 2 (UFD-2) ubiquitin ligase (Ackermann et al., 2016). To determine whether the presence of RAD-51–free ssDNA in *top-3* mutants could trigger DNA damage–induced apoptosis, we quantified the apoptotic nuclei in *top-3; rad-51^RNAi*. We observed that the apoptosis levels in *top-3; rad-51^RNAi* did not differ from those of *rad-51^RNAi*, indicating that the apoptotic defects of *top-3* could be overcome in this genetic background (Fig. 4 A). In *top-3*, RAD-51 foci were loaded onto stalled replication forks in a *rfs-1*–dependent manner in the mitotic zone (Fig. 1 G and Fig. S2 A; *rfs-1* encodes a RAD-51 paralog). To examine if the specific removal of RAD-51 in the mitotic zone would alleviate the apoptosis block, we quantified apoptosis in the double mutant *top-3 rfs-1*. Indeed, RAD-51 removal from mitotic recombination intermediates increased the apoptosis levels (Fig. 4 B). Moreover, we confirmed with *spo-11^RNAi* that these "apoptosis-inducing" recombination intermediates occurred independently of meiotic DSBs (Fig. 4 C). Collectively, we conclude that the apoptosis block in *top-3* can be relieved by the removal of RAD-51 from the mitotic repair intermediates.

To better understand the specific stage when the RAD-51–free ssDNA would be susceptible to apoptosis induction, we generated a conditional *rad-51* mutant by insertion of an auxin-inducible degron fused to the RAD-51 protein (Fig. S3 A for functionality of the *rad-51::degron* and efficiency of degradation). Meiocytes in the gonad represent a spatiotemporal gradient of meiotic prophase I, with nuclei moving down the gonad tube with a speed of approximately one cell row/hour (Jaramillo-Lambert et al., 2007). 5-ethynyl-2′-deoxyuridine (EdU) incorporation showed that the flux of nuclei through the gonad in *top-3* did not differ from WT, despite an extended mitotic region in *top-3* (Fig. S3 B).

We then induced RAD-51 degradation to establish a time course of depletion in WT and *top-3* germlines. Since it takes ~40 h for nuclei to move from the mitosis/meiosis transition zone to the death zone (the gonad region competent for apoptosis), we examined apoptosis at 20, 30, and 40 h after inducing RAD-51 degradation, which correspond to mid-pachynema, early pachynema, and mitosis/meiosis transition, respectively (Fig. 4, D and E). In the WT, RAD-51 degradation in mid-pachynema already led to apoptosis induction (Fig. 4 E, left). Strikingly, in *top-3*, apoptosis was elevated only at 40 h after auxin treatment (Fig. 4 E, right), indicating that the lesions that become competent to apoptosis induction reside at the mitosis/meiosis transition. Although removal of RAD-51 from resected ssDNA normally leads to apoptosis up-regulation in the WT (Fig. 4 A and Fig. 4 E, left; Alpi et al., 2003), we did not observe this after 20–30 h of auxin exposure in *top-3* (Fig. 4 E, right); under these conditions, RAD-51 should have been removed from the *spo-11*–induced DSBs.

Altogether, this suggests that the RAD-51–bound recombination intermediates originating in the premeiotic zone can

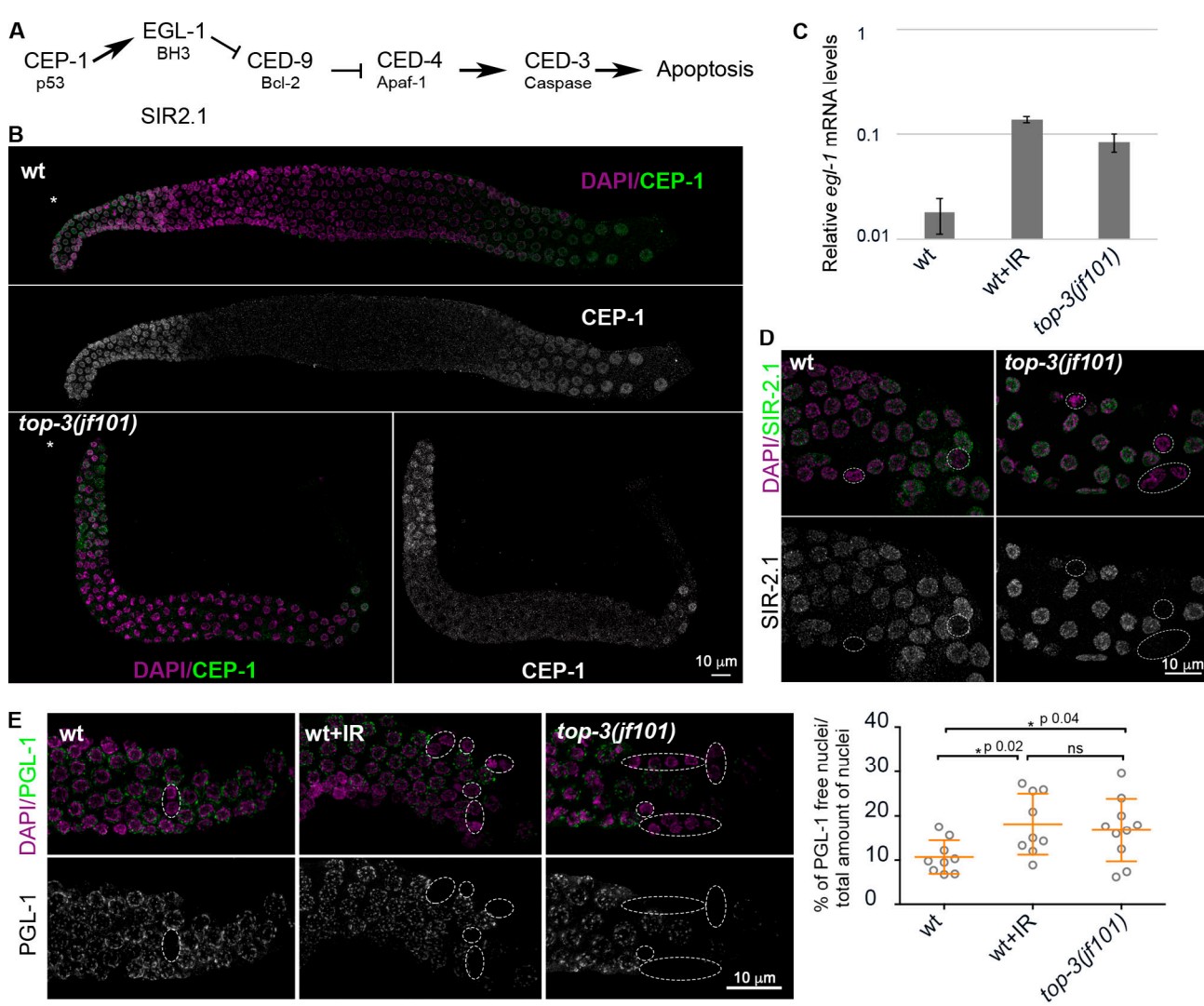

Figure 3. **The apoptotic machinery is properly activated in *top-3*. (A)** Schematic representation of the *C. elegans* apoptosis pathway. **(B)** Representative images of gonads stained with DAPI (magenta) and anti-CEP-1 antibody (green) in the wt and *top-3(jf101)*. *, Mitotic tip. **(C)** Relative levels of *egl-1* mRNA, as assessed by RT-qPCR, in the wt, wt irradiated with 120 Gy IR, and *top-3(jf101)*. Error bars indicate the SD (also see Fig. S2 F). **(D)** SIR-2.1 localization in late pachynema in wt and *top-3(jf101)* gonads stained with DAPI (magenta) and anti-SIR-2.1 antibody (green). Dashed circles indicate nuclei without SIR-2.1 staining. **(E)** Left: PGL-1 localization in late pachynema in the indicated genotypes. Gonads were stained with DAPI (magenta) and anti-PGL-1 antibody (green). Dashed circles indicate nuclei without PGL-1 staining. Right: Percentage of nuclei without PGL-1 staining in the last 10 cell rows of the gonad (death zone), the zone competent for apoptosis execution. At least nine gonads/genotype were assessed. wt worms were exposed to 120 Gy. Error bars indicate the mean ± SD: wt, 10.7 ± 3.8; wt + IR, 18.1 ± 6.9; and *top-3(jf101)*, 16.8 ± 7. P values were calculated using the Mann–Whitney test. wt, wild-type.

augment apoptosis upon RAD-51 removal. However, this does not hold true for meiotic breaks because RAD-51 depletion from meiotic recombination intermediates in *top-3* did not trigger enhanced apoptosis. Therefore, meiotic breaks are probably processed aberrantly at a step before RAD-51 loading.

To determine whether (1) the DSBs in meiotic prophase are unable to induce efficient apoptosis due to the "chromatin pathologies" established in the premeiotic stage or (2) the lack of TOP-3 in meiosis directly affects them, we generated an internally tagged *top-3::ha::degron* line to specifically deplete TOP-3 in meiosis to examine apoptosis under meiotic deletion of TOP-3 (Fig. S3 C for functionality of the *top-3::ha::degron* and efficiency of degradation; TIR1 is driven by the *sun-1* promoter, which drives expression in all cells in the germline; Zhang et al., 2015).

We confirmed depletion of TOP-3 by analyzing protein abundance in the gonad. TOP-3 localizes as distinct foci throughout pachynema (Fig. 5 A) with similar dynamics as has been seen for RMH-1 and HIM-6 foci (Jagut et al., 2016). TOP-3–depleted pachytene cells had a significantly less efficient apoptotic response after irradiation compared with *top-3::ha::degron* worms without auxin treatment (Fig. 5 B). Therefore, we conclude that lack of TOP-3 generates DNA lesions in both premeiotic and meiotic germline compartments that are less capable of inducing a massive apoptotic response.

## Depletion of NHEJ, Alt-NHEJ, or SSA increases apoptosis in *top-3* mutants

Since in *top-3* mutants the apoptosis block is alleviated only when RAD-51 is removed from the mitotic DNA intermediates,

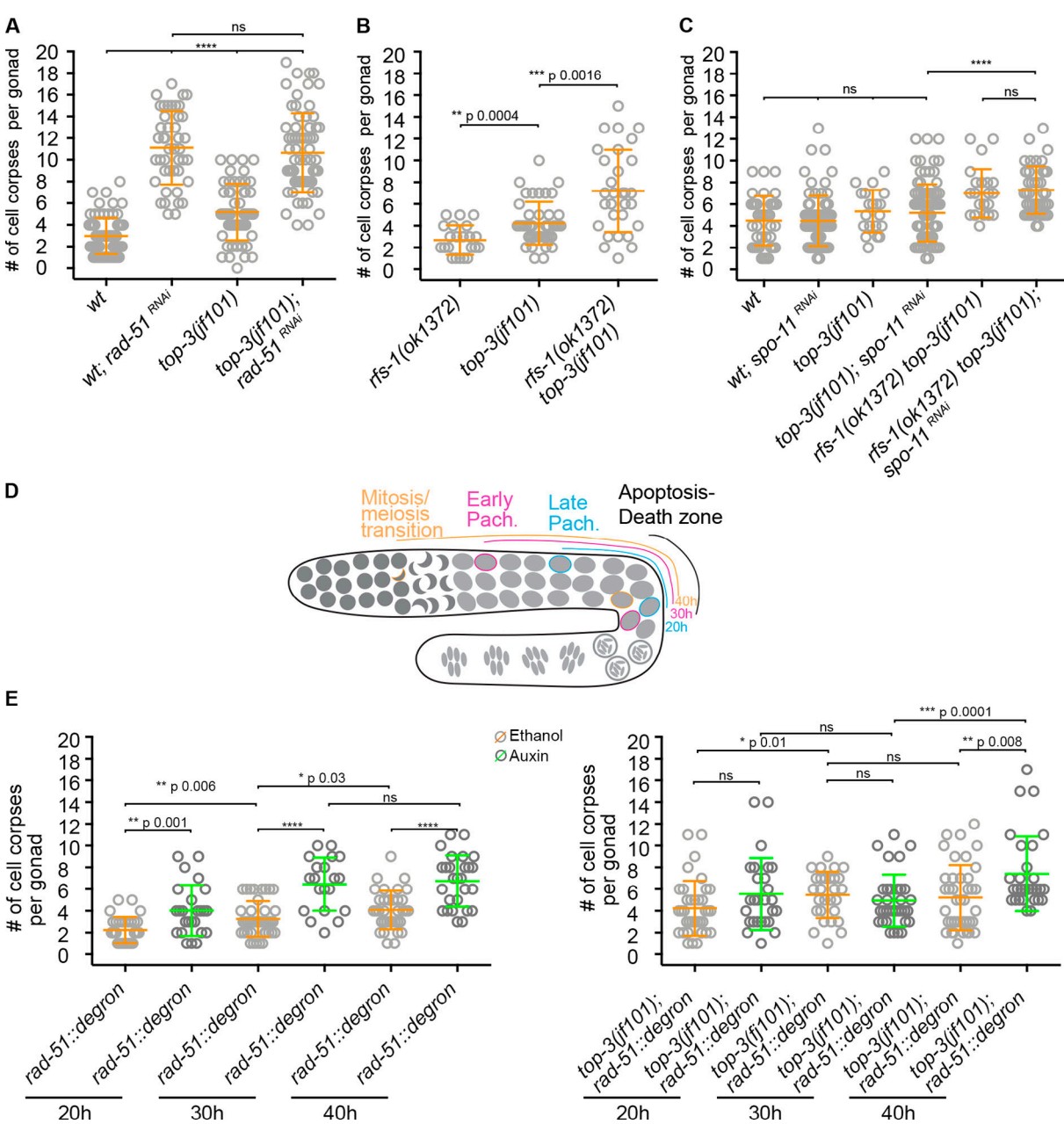

Figure 4. **RAD-51 depletion in premeiotic nuclei triggers DNA damage apoptosis in the *top-3* mutant. (A–C)** Apoptosis quantification using SYTO-12 in the indicated genotypes. Scatter plots indicate the mean ± SD; *n* = number of gonads scored. **(A)** wt and *top-3(jf101)* subjected to *rad-51*[RNAi]: wt, 3 ± 1.7, *n* = 76; wt; *rad-51*[RNAi], 11.1 ± 3.4, *n* = 44; *top-3(jf101)*, 5.2 ± 2.6, *n* = 54; and *top-3(jf101)*; *rad-51*[RNAi], 10.65 ± 3.7, *n* = 68. ****, P < 0.0001, ns , calculated using the Mann–Whitney test. **(B)** *rfs-1(ok1372)*, 2.7 ± 1.4, *n* = 22; *top-3(jf101)* (quantification as described in Fig. 2 D), 4.2 ± 1.9, *n* = 43; and *rfs-1(ok1372) top-3(jf101)*, 7.2 ± 3.8, *n* = 30. P values were calculated using the Mann–Whitney test. **(C)** wt, 4.5 ± 2.3, *n* = 41; wt; *spo-11*[RNAi], 4.5 ± 2.3, *n* = 78; *top-3(jf101)*, 5.3 ± 1.9, *n* = 20; *top-3(jf101)*; *spo-11*[RNAi], 5.2 ± 2.6, *n* = 89; *rfs-1(ok1372) top-3(jf101)*, 7 ± 2.2, *n* = 24; and *rfs-1(ok1372) top-3(jf101)*; and *spo-11*[RNAi], 7.3 ± 2.2, *n* = 44. ****, P < 0.0001, ns, calculated using the Mann–Whitney test. **(D)** Schematic representation of RAD-51 depletion in the *C. elegans* germline. The indicated number of hours of exposure to auxin or solvent (ethanol) corresponds to the periods of exposure during prophase before apoptosis in pachynema (death zone; e.g., exposed for 20 h means that the nuclei were in mid-pachynema when RAD-51 was depleted; also see Fig. S3 A for RAD-51 depletion upon auxin treatment). Pach., pachynema. **(E)** Time-course analysis of the *rad-51::degron* control (left) and *top-3(jf101)*; *rad-51::degron* (right); apoptosis was quantified with SYTO-12 in the absence (orange) or presence (green) of auxin. *n* = number of gonads scored. *rad-51::degron* (ethanol: 20 hours (h), 2.3 ± 1.2, *n* = 32; 30 h, 3.3 ± 1.6, *n* = 41; 40 h, 4.1 ± 1.8, *n* = 35; auxin: 20 h, 4 ± 2.3, *n* = 25; 30 h, 6.5 ± 2.4, *n* = 20; 40 h, 6.7 ± 2.4, *n* = 27); *top-3(jf101)*; and *rad-51::degron* (ethanol: 20 h, 4.2 ± 2.5, *n* = 37; 30 h, 5.5 ± 2.1, *n* = 28; 40 h, 5.2 ± 3, *n* = 40; auxin: 20 h, 5.6 ± 3.3, *n* = 27; 30 h, 4.9 ± 2.4, *n* = 34; 40 h, 7.4 ± 3.5, *n* = 29). Error bars indicate the mean (and SD), ****, P < 0.0001; ns, calculated using the Mann–Whitney test.

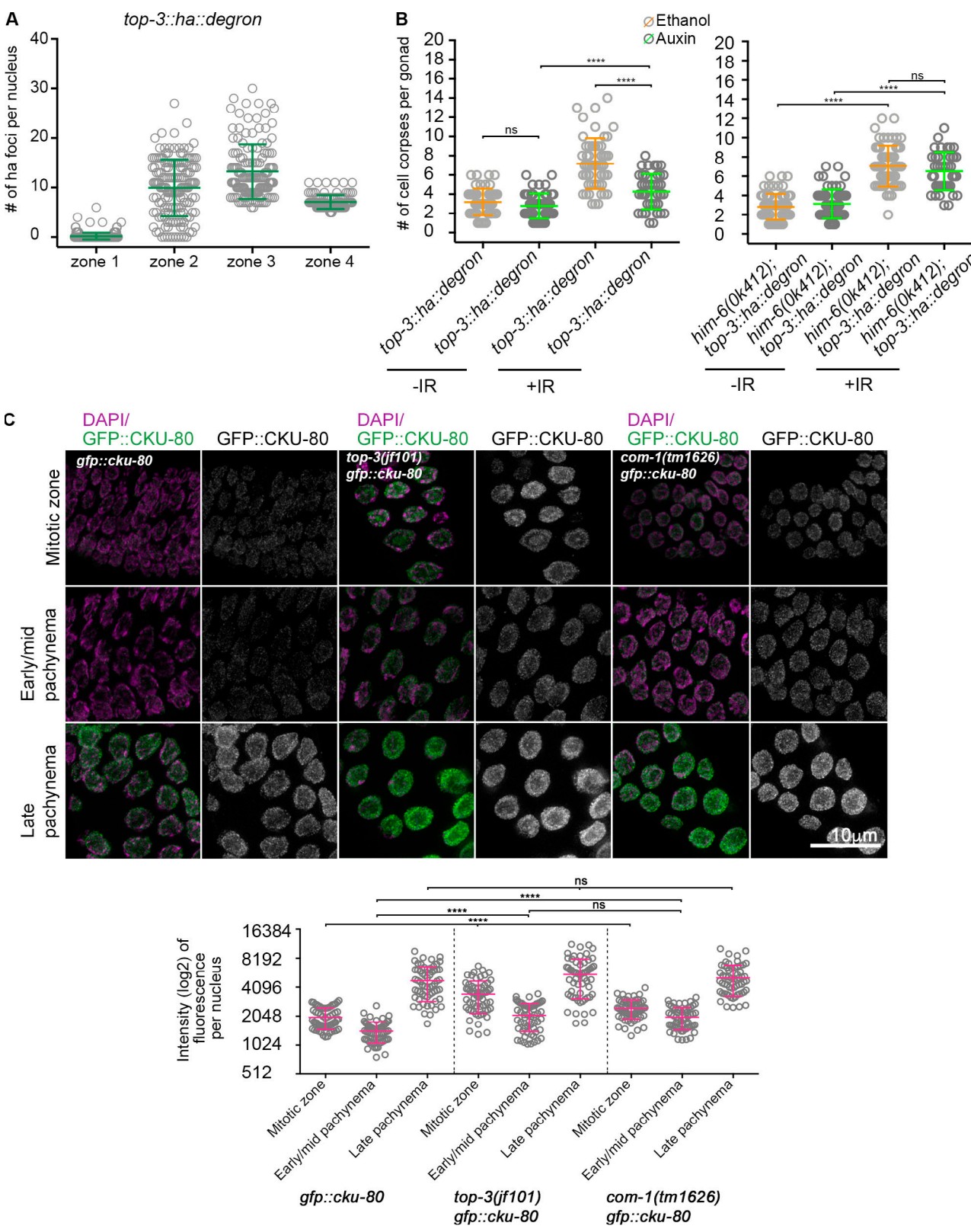

Figure 5. **Meiotic depletion of TOP-3 down-regulates apoptosis and up-regulates CKU-80 in *top-3*. (A)** Quantification of TOP-3::HA foci in the *C. elegans* meiotic part of the gonad divided in four equal zones; the scatter plot indicates the number of foci per nucleus. For each zone, *n* = number of nuclei quantified; zone 1, 0.2 ± 0.7, *n* = 218; zone 2, 9.9 ± 5.7, *n* = 156; zone 3, 13.2 ± 5.5, *n* = 142; zone 4, 7.1 ± 1.4, *n* = 107. Error bars indicate the mean (and SD). **(B)** Apoptosis quantification in *top-3::ha::degron* (left) and *him-6(ok412); top-3::ha::degron* (right) in the absence (orange) or presence (green) of auxin using SYTO-12. *n* = number of gonads scored. *top-3::ha::degron* (not irradiated) ethanol, 3.2 ± 1.4, *n* = 50; auxin, 2.8 ± 1.3, *n* = 51; and *top-3::ha::degron* (after irradiation) ethanol 7.2 ± 2.6, *n* = 54; auxin 4.3 ± 1.8, *n* = 44. *him-6(ok412); top-3::ha::degron* not irradiated: ethanol, 2.8 ± 1.3, *n* = 77; and auxin, 3.1 ± 1.5, *n* = 60. *him-6(ok412); top-3::*

ha::degron after irradiation: ethanol, 7.1 ± 2.1, n = 59; and auxin, 6.5 ± 2, n = 41. Error bars indicate the mean (and SD), ****, P < 0.0001; ns, calculated using the Mann–Whitney test. **(C)** Top: CKU-80 localization during prophase I in *gfp::cku-80*, *top-3(jf101) gfp::cku-80* and *com-1(tm1626) gfp::cku-80* gonads (also see Fig. S4 A). Gonads were stained with DAPI (magenta), and the GFP::CKU-80 fluorescence signal was recorded (green). Bottom: Quantification of the fluorescence signal during different prophase I stages in the indicated mutants. The scatter plot shows the mean ± SD. The y axis is in log2 scale to better highlight the distribution of the numbers. ****, P < 0.0001, calculated using the Mann–Whitney test. *gfp::cku-80*, 1995–1428–4786; *top-3(jf101) gfp::cku-80*, 3465–2092–5557; and *com-1(tm1626) gfp::cku-80*, 2468–1998–5110.

we wanted to examine if alternative repair pathways, which come into play before RAD-51 loading and when HR is compromised, could alleviate the apoptosis block in meiosis. In *C. elegans*, CKU-80 has been observed to bind early recombination intermediates when processing to RAD-51–loaded filaments is compromised (Lemmens et al., 2013). Therefore, we generated a functional GFP-tagged CKU-80 line (Fig. S4 A) and analyzed its CKU-80 expression/loading dynamics.

Strikingly, we detected CKU-80 accumulation in the mitotic gonad compartment and early/mid-pachynema in *top-3* (Fig. 5 C); in contrast, CKU-80 was present at lower levels in the WT. For comparison, we measured CKU-80 accumulation in the resection mutant, *com-1* (Fig. 5 C). CKU-80 levels were comparable in early/mid-pachynema in *top-3* and *com-1* but not in the mitotic compartment, indicating that *top-3* may encounter problems at the early stages of meiotic DNA repair, similar to *com-1*.

We therefore generated *cku-80 top-3* double mutants and found a remarkable increase in apoptosis (Fig. 6 A), together with *egl-1* transcription levels comparable to the *top-3* single mutant (Fig. S2 F). The double mutant did not display increased numbers of germline nuclei (Fig. S2 G), and EdU incorporation experiments did not suggest accelerated meiotic progression (Fig. S3 B). This result rules out the possibility that either changes in nuclei progression through the gonad or differing numbers of nuclei in the gonad account for the apoptosis increase in the double mutant. Similarly, we detected a significant increase in the apoptosis levels in *cku-80 top-3::ha::degron* upon IR after meiotic depletion of *top-3* (Fig. S4, B and C). To find out whether activation of the NHEJ pathway dampens the apoptotic response in *top-3*, we also analyzed apoptosis in the double mutants *top-3 cku-70*, and *top-3 lig-4* (Fig. 6 A). Apoptosis induction was similar in *top-3 cku-70* and *top-3 cku-80*, but slightly lower in *top-3 lig-4*. Interestingly, the *top-3*–dependent apoptosis block was overcome in the *top-3 polq-1* double mutant (defective in Alt-NHEJ; Fig. 6 A). Also, *top-3 polq-1* did not display increased numbers of germline nuclei (Fig. S2 G). Concurrent impairment of NHEJ and Alt-NHEJ in *top-3* did not display an additive effect (Fig. 6 A). These results suggest that both NHEJ and Alt-NHEJ affect the efficiency of apoptosis execution in *top-3*.

Since repair via SSA resembles Alt-NHEJ, we tested whether abrogation of SSA would similarly increase the apoptosis levels. We therefore depleted the endonuclease XPF-1 acting in SSA in the *top-3* mutant. Indeed, we found a similar increase in the apoptotic levels in the *xpf-1; top-3* double mutant (Fig. 6 A). XPF-1 not only is employed in the SSA pathway but also functions as a resolvase in *C. elegans* (Agostinho et al., 2013; Saito et al., 2009). We therefore examined whether deficiency in another resolution enzyme, like the endonuclease MUS-81, would increase apoptosis in *top-3* mutants. Strikingly, *mus-81; top-3* displayed

apoptotic levels comparable to *top-3* single mutants (Fig. S4 D), implying that interfering with resolution does not enhance the apoptotic levels per se. Thus, it is likely that the intermediates that interfere with the apoptosis signaling are generated at an early stage during the DSBs repair.

We suggest that in *top-3* mutants, aberrant recombination intermediates activate the NHEJ, Alt-NHEJ, and SSA pathways and that this affects the apoptotic response. The lack of fragmentation of diakinesis chromosomes in *top-3 polq-1 cku-70* (Fig. S1 C) does not allow us to reach a definite conclusion about to what extent DNA repair occurs via those pathways. The presence of complex joint DNA structures in *top-3* precludes this analysis (also see Fig. S1, B–D; and Discussion).

As CKU-80 can accumulate at both premeiotic and meiotic DNA breaks, we investigated how much meiotic DNA lesions contribute to apoptosis induction by CKU-80 depletion. Strikingly, germline apoptosis levels did not differ between *top-3 cku-80; spo-11^{RNAi}* and the *top-3* single mutant (Fig. 6 B). This result suggests that in the *top-3* mutant, meiotic DNA breaks might be largely repaired by noncanonical alternative pathway(s), contributing to the low apoptotic response. Furthermore, our evidence shows that in *top-3*, apoptosis triggered by CKU-80 removal is *cep-1*–dependent (Fig. 6 C).

### Depletion of Bloom helicase HIM-6 increases apoptosis in *top-3* mutants

To gain further insight into the nature of the aberrant recombination intermediates that are largely refractory to HR, we generated a *him-6; top-3::ha::degron* strain and analyzed apoptosis after meiotic depletion of TOP-3. It has been previously shown that the double mutant *him-6; top-3* induces a mitotic catastrophe (Wicky et al., 2004), which precludes the analysis of the effect of HIM-6 on recombination intermediates generated in the mitotic gonad zone. Remarkably, meiotic depletion of both TOP-3 and HIM-6 showed an apoptotic response equally high as the one we observed in the *him-6* single mutant (Fig. 5 B, Fig. S4 C, and Fig. 2 E). This strongly suggests that in the absence of the topoisomerase, the Bloom helicase (HIM-6) creates aberrant recombination intermediates that are less capable to use HR for their repair in meiocytes.

## Discussion

The activities of the STR/BTR complex in meiosis have been meticulously worked out in *Saccharomyces cerevisiae* (for a review, see Haber, 2015). The defects relate to failure to reject invaded D-loops and to dissolve/decatenate joint DNA structures. An as yet poorly understood CO activity has also been reported, in particular in *C. elegans* (Jagut et al., 2016; Schvarzstein

Figure 6. **Impairment of the NHEJ/Alt-NHEJ/SSA pathways in meiosis triggers p53/CEP-1-dependent apoptosis. (A)** Apoptosis quantification using SYTO-12 in the indicated genotypes. $n$ = number of gonads scored. *top-3(jf101)*, 4.2 ± 1.9, $n$ = 43 (quantification as in Fig. 2 D); *cku-80(ok861)*, 2.9 ± 1.3, $n$ = 65;

*cku-80(ok861) top-3(jf101)*, 8.4 ± 3, n = 47; *cku-70(tm1524)*, 2.2 ± 1.3, n = 24; *cku-70(tm1524) top-3(jf101)*, 8.2 ± 2.6, n = 52; *lig-4(ok716)*, 2.2 ± 1, n = 52; *top-3(jf101) lig-4(ok716)*, 7.2 ± 2.6, n = 98; *polq-1(tm2026)*, 2.3 ± 1.1, n = 65; *polq-1(tm2026) top-3(jf101)*, 6 ± 2.1, n = 70; *cku-70(tm1524) polq-1(tm2026)*, 3.2 ± 1.6, n = 80; and *top-3(jf101) cku-70(tm1524) polq-1(tm2026)*, 7.3 ± 2.8, n = 83; *xpf-1(tm2842)*, 4.1 ± 1.7, n = 52; *top-3(jf101); xpf-1(tm2842)*, 6.7 ± 3.4, n = 56. ****, P < 0.0001; ns,calculated using the Mann–Whitney test. **(B)** Apoptosis quantification with SYTO-12 in the indicated genotypes (quantification as described for Fig. 4 C for wt, wt; *spo-11^RNAi*; *top-3(jf101)* and *top-3(jf101); spo-11^RNAi*): n = number of gonads scored: wt, 4.5 ± 2.3, n = 41; wt; *spo-11^RNAi*, 4.5 ± 2.3, n = 78; *top-3(jf101)*, 5.3 ± 1.9, n = 20; *top-3(jf101); spo-11^RNAi*, 5.2 ± 2.6, n = 89; *top-3(jf101) cku-80(ok861)*, 9.2 ± 2.5, n = 18; and *top-3(jf101) cku-80(ok861); spo-11^RNAi*, 4.8 ± 2, n = 88. ****, P < 0.0001; ns, calculated using the Mann–Whitney test. Error bars indicate mean ± SD. **(C)** Apoptosis quantification with SYTO-12 in the indicated genotypes. n = number of gonads scored: *cep-1(gk138)*, 2.1 ± 1.2, n = 67; *top-3(jf101)*, 4.2 ± 2, n = 43; *cep-1(gk138); top-3(jf101)*, 1.9 ± 1, n = 66; *cku-80(ok861) top-3(jf101)*, 8.5 ± 3.5, n = 25; and *cep-1(gk138); top-3(jf101) cku-80(ok861)*, 1.9 ± 1.2, n = 58. Error bars indicate the mean (and SD), ****, P < 0.0001 calculated using the Mann–Whitney test. **(D)** Model showing the function of topoisomerase 3 during mitosis and meiosis (green to blue gradient). In mitosis, TOP-3 is involved in processing stalled and collapsed replication forks; in meiosis, it interferes with strand invasion and is involved in dissolving joint molecules. In mitosis, *top-3* mutants accumulate aberrant recombination intermediates that are probably coated with RAD-51. Those abnormal intermediates are imported into meiosis, where they are mostly refractory to apoptosis. In meiosis, HIM-6 generates aberrant intermediates that are directed toward repair via the NHEJ, Alt-NHEJ, and SSA pathways in the *top-3* mutant, which prevents efficient apoptosis. wt, wild-type.

et al., 2014; Wicky et al., 2004). The STR/BTR complex is not reported to make a major contribution to the resection steps in meiosis (for review, see Haber, 2015). Based on our findings, we showed a previously undescribed consequence of mutating topoisomerase 3 in the germline. Elimination of topoisomerase 3 function creates aberrant recombination intermediates in the premeiotic and meiotic compartments of the gonad. However, these do not lead to efficient *cep-1*/p53-dependent germline apoptosis (model shown in Fig. 6 D). The *top-3* mutant probably accumulates replicative lesions, onto which RAD-51 is continuously loaded throughout meiosis. The RAD-51 paralog, RFS-1, acts on these lesions by promoting RAD-51 recruitment (Ward et al., 2007), and *rfs-1* depletion renders these lesions competent to trigger apoptosis. This is consistent with our observation that RAD-51 depletion in premeiotic nuclei also increases apoptosis in *top-3* mutants. Exclusive meiotic depletion of RAD-51 and TOP-3 showed that recombination intermediates in meiocytes are also less effective in triggering apoptosis, probably via an intermediate upstream of RAD-51 loading.

Consistent with CKU-80 up-regulation in *top-3* mutants, aberrant recombination intermediates caused by the absence of TOP-3 might direct meiotic breaks away from repair via HR toward normally not employed repair pathways. With this, recombination intermediates are no longer detected as unrepaired DNA lesions, and the nuclei evade culling by apoptosis. To test whether TOP-3 has a more direct role in apoptosis itself, we have also performed an experiment with *top-3::ha::degron* degrading TOP-3 at a later stage during recombination (data not shown). We could observe that late degradation of TOP-3 several hours after irradiation had no effect on the apoptotic response, which was as high as in the auxin untreated control. This also supports a model where the recombination intermediate per se causes the unexpected low apoptosis in *top-3*.

The fact that apoptosis levels in *top-3* mutants can be increased by mutating the classical NEHJ, Alt-NHEJ, or SSA pathways strongly suggests that defective recombination intermediates direct DNA repair toward these pathways, preventing a critical level of a key apoptosis factor(s) from being reached. *egl-1* transcription is up-regulated in *top-3* to comparable levels to those observed in irradiated WT. Our analysis of the landmark events of DNA damage–induced apoptosis suggests that apoptosis is initiated but not fully executed in the *top-3* mutant, and that physiological apoptosis levels are not affected.

The processing of the aberrant lesions by these alternative pathways does not affect chromosome configurations at diakinesis, since *top-3 polq-1 cku-70* mutants do not display increased chromosome fragmentation as a consequence of unrepaired meiotic breaks. However, the disorganized, uncondensed, and highly abnormal diakinesis DAPI bodies in *top-3* persist in the absence of several repair pathways (e.g., in the *top-3; zhp-3* double mutant, where we would expect a lot of univalents; Fig. S1 D); therefore, we assume that the decatenation of certain joint structures might actually rely on TOP-3 activity. Premeiotic DNA lesions occurring in *top-3* mutants progressively accumulate RAD-51 signals (even in the absence of SPO-11–induced DNA breaks), possibly creating a "pathological chromatin" that reinforces the joint structures.

An alternative explanation for the reduced apoptosis levels in *top-3* mutants is that CKU-80 persists at the ends of the DNA break sites (as in *com-1* mutants; Lemmens et al., 2013), thereby inhibiting resection. However, the facts that apoptosis is also boosted in the *top-3 polq-1* double mutant or RPA-1–coated ssDNA accumulates in the *top-3* mutant argue against this possibility.

Remarkably, *rmh-1*, *rmh-2*, and *him-6* single mutants mount an effective apoptotic response after irradiation. Dna2-mediated DNA resection in mitosis is supported jointly by the helicase and the topoisomerase (Bizard and Hickson, 2014). The *him-6* helicase mutant is proficient in the apoptotic response to irradiation and displays CKU-80 expression similar to the GFP::CKU-80 (Fig. S4 A). Thus, this might argue against a model in which the absence of HIM-6 alone would generate faulty resected intermediates that prefer non-HR DNA repair pathways. In meiocytes, we observed that the elimination of the helicase alleviates the apoptosis block in *top-3* mutants. This might indicate that aberrant activities of the helicase render recombination intermediates more prone to use alternative repair pathways, when TOP-3 is not available.

We do not know the exact nature of the aberrant recombination intermediates, but in meiosis, they seem to occur upstream of *rad-51* (Fig. 4 E) and could still have to do with controlled resection. Our genetic analysis proposes that the unscheduled meiotic activities of HIM-6 affecting those recombination intermediates are normally counteracted by TOP-3.

In summary, aberrant recombination intermediates in *top-3* mutant *C. elegans* meiocytes remarkably have a severe impact on

healthy oocyte production by undermining the efficient elimination of faulty oocytes by apoptosis.

## Materials and methods

### Experimental model and strain details

Worm strains were maintained at 20°C (Brenner, 1974) and, unless otherwise indicated, on nematode growth medium (NGM) agar plates seeded with *Escherichia coli* OP50. Hermaphrodites were used in all experiments. The N2 Bristol strain was used as the WT control. See Table S1 for a complete list of strains, reagents, and resources.

### Cytology

Immunofluorescence analysis of worms at 20–24 h after L4 stage was performed as previously described (Martinez-Perez and Villeneuve, 2005). Gonads were dissected in 1× PBS on poly-L-lysine–coated slides, fixed with 1% paraformaldehyde in 1× PBS for 5 min at room temperature, and then frozen in liquid nitrogen. After freeze-cracking, slides were incubated in methanol at –20°C for 1 min, before washing three times in PBS containing 1% Tween-20 (1× PBST) for 5 min at room temperature. Nonspecific binding sites were blocked by incubation in 1% BSA in 1× PBST for at least 30 min. Primary antibodies diluted in 1× PBST were applied and incubated overnight at 4°C. Slides were then washed three times in 1× PBST at room temperature, and secondary antibodies were applied for 2 h. After three washes in PBST, 2 μg/ml DAPI was applied for 1 min, and slides were then washed for at least 20 min in 1× PBST. Slides were mounted with Vectashield.

[RPA-1::YFP] was analyzed in dissected gonads after fixation in 2.5% paraformaldehyde for 1 min and freezing in liquid nitrogen. After the freeze crack, the slides were incubated in 95% ethanol for 1 min as previously described (Checchi et al., 2014).

For GFP::CKU-80 detection, worms were dissected in egg buffer containing 0.1% Tween-20 and directly frozen in liquid nitrogen. After freeze-cracking, slides were incubated in methanol at –20°C for 1 min. Gonads were fixed with 4% PFA in 100 mM $K_2HPO_4$, washed three times in 1× PBST, and incubated with DAPI. For visualization of CED-1::GFP (Fig. 2 C), worms were fixed in pure ethanol before applying DAPI and Vectashield.

The following primary antibodies were used: rabbit anti-RAD-51 (1:1,000 dilution; a gift from M. Zetka, McGill University, Montreal, Canada), rabbit anti-pCDK-1 (1:150 dilution; Calbiochem; cat. #219440), sheep anti-CEP-1 (1:100 dilution; a gift from A. Gartner, Ulsan National Institute of Science and Technology: Ulsan, Ulsan, Korea), goat anti-SIR2.1 (1:100 dilution; a gift from A. Gartner; Greiss et al., 2008), mouse anti-PGL-1 (1 μg/μl; a gift from T. Hyman, Max Planck Institute of Molecular Biology and Genetics, Dresden, Germany), rabbit anti-HA (preabsorbed to WT worms, 1:100 dilution; Sigma-Aldrich; cat. #H6908), rhodamine-conjugated sheep anti-digoxigenin (1:100 dilution; Roche; cat. #11207750910), and FITC-conjugated goat anti-biotin (1:250 dilution; Abcam; cat. #ab6650). Appropriate secondary antibodies conjugated to Alexa Fluor 488 or 594 were used at a dilution of 1:500.

Gonads were dissected from worms at 20–24 h after L4 stage to quantify DAPI bodies at diakinesis and the total number of germlines nuclei. For the antibodies used, image acquisition and quantification of signal, see below.

### Quantification and kinetics of RAD51 foci

RAD-51 quantification (Fig. S2 A) was performed by dividing the gonad into seven equal zones from the distal tip to late pachynema, and counting the number of foci/nucleus for each zone. Graphs were used to display the percentage of nuclei in the following categories: 0 foci, 1 focus, 2 or 3 foci, 4–6 foci, 7–12 foci, >12 foci, and stretches (a continuous signal not resolved into foci). Three gonads were assessed per genotype.

### Quantification of CKU-80 signals

GFP::CKU-80 signals were quantified by measuring the integrated intensity fluorescence signal in encircled nuclei normalized to the measured area (after subtraction of the WT background signal; Fig. 5 C) with Fiji ImageJ software. Germline images were acquired using the same settings, deconvolved, and projected before the analysis. 10 nuclei from each zone were randomly chosen for quantification. Six gonads were quantified per genotype.

### Quantification of TOP-3 foci

TOP-3 quantification was performed by dividing the gonad into four equal zones from the transition zone to late pachynema and counting the number of foci/nucleus for each zone. Three gonads were assessed per genotype.

### Image acquisition

Images were acquired at room temperature using a Delta Vision system equipped with 60×/1.42 and 100×/1.40 oil immersion objective lenses and a cooled charge-coupled device camera, an Ultra Delta Vision microscope equipped with UPlanXApo 60x/1.42 Oil and UPlanSApo 100x/1.4 Oil objective and a 4-megapixel sCMOS camera. Z stacks of 0.20 μm were deconvolved using SoftWoRx software and processed in Adobe Photoshop. Three to five images were acquired using the same settings to cover each gonad. Maximum intensity projections of deconvolved images were generated using Fiji/ImageJ after background subtraction using a rolling-ball radius of 50 pixels. Where specified, images of gonads consist of multiple stitched images; this was necessary due to the size limitation of the field of view at high magnifications. Stitching of images to build up entire gonads was performed manually in Adobe Photoshop. Levels of stitched images were adjusted relative to each other in Adobe Photoshop to correct for auto-adjustment settings of the microscope, as previously described (Link et al., 2018).

### Apoptosis quantification

To quantify apoptotic nuclei after irradiation, worms (20 h after L4 stage) were irradiated using a [137]Cs source and scored after 20 h (7 h for *glp-1* mutants). Worms were soaked in 33 μM SYTO-12 in M9 buffer for 2 h and then transferred to seeded NGM plates for at least 20 min before being mounted onto a 2% agarose pad for quantification. For CED-1::GFP assessment, worms at 20–24 h after L4 stage were mounted onto 2% agarose pads, and apoptotic nuclei were quantified by scoring the GFP signal

in the engulfing cells. The number of positive nuclei per gonad was scored using a Zeiss Axio microscope equipped with a 40× or 63× oil immersion objective lens.

### RNAi

RNAi was administered using the feeding or soaking methods (Kamath and Ahringer, 2003).

*rad-51* and *cku-80* RNAi was performed using Ahringer library clones. *spo-11* RNAi was performed by soaking worms with dsRNA (nucleotides 593–1,500 in cosmid T05E11). Bacteria (HT115) expressing *rad-51, cku-80* dsRNA, or pL4440 empty vector (control) were streaked onto plates containing 100 mg/ml ampicillin and 12.5 mg/ml tetracycline and incubated overnight at 37°C. Single colonies were then incubated overnight at 37°C in liquid Luria-Bertani (LB) medium containing 100 mg/ml ampicillin. Liquid cultures were centrifuged (2,000 *g*, 20 min at 4°C), and bacterial pellets were resuspended in 1.5–2 ml medium (as described above) supplemented with 1 mM IPTG + ampicillin, plated into NGM plates, and incubated at 37°C overnight to induce dsRNA expression. L4-stage worms were then transferred to *RNAi* plates. The efficiency of *rad-51 RNAi* was verified by RAD-51 immunostaining and of *cku-80 RNAi* by rescue of the diakinesis phenotype of the *com-1* mutant (Lemmens et al., 2013).

*spo-11 RNAi* was performed by soaking worms with dsRNA (corresponding to nucleotides 593–1,500 of the T05E11 cosmid). Templates were generated by PCR using the following primer pairs containing the T3 promoter (indicated by lower case letters): (1) forward: 5′-aattaaccctcactaaaggGTCGGGCTAATTTAAACATT-3′, reverse: 5′-TGAATCTTCGTGGTACTCGT-3′; and (2) forward: 5′-aattaaccctcactaaaggTGAATCTTCGTGGTACTCGT-3′; reverse: 5′-GTCGGGCTAATTTAAACATT-3′.

Purified DNA (0.1–0.2 µg) was used for in vitro transcription using a MEGAscript T3 Transcription Kit (Ambion) according to the manufacturer's instructions. Total RNA was isolated using TRIzol (TriFast Peqlab; VWR) according to the manufacturer's protocol and quantified using a Nanodrop spectrophotometer. To anneal the dsRNA, equal amounts (3,000 ng/µl) of the two ssRNAs were incubated in 3× soaking buffer (32.7 mM Na$_2$HPO$_4$, 16.5 mM KH$_2$PO$_4$, 6.3 mM NaCl, and 14.1 mM NH$_4$Cl) at 68°C for 10 min and then at 37°C for 30 min. L4-stage worms were then soaked in dsRNA at a concentration of 1 µg/µl for 24 h and transferred onto NGM plates; apoptotic nuclei were scored 48 h later. Efficient *spo-11 RNAi* was indicated by the presence of 12 univalents in diakinesis nuclei as indicated in Fig. S4 E.

### FISH

FISH was performed as previously described (Silva et al., 2014). Gonads of young adult worms were dissected and fixed in 7.4% paraformaldehyde. After washing three times in 2× SCCT (1x saline-sodium citrate buffer [SSC] with 0.1% Tween-20) buffer, gonads were dehydrated by incubation in increasing ethanol concentrations and air-dried at room temperature. Chromosome III containing the pairing center (cosmid T17A3) and the chromosome V 5S ribosomal DNA (rDNA) locus labeled with biotin and digoxigenin, respectively, were used as FISH probes according to the manufacturer's instructions for the Nick

Translation Kit (Sigma-Aldrich). Denaturation and annealing were performed on a heat block for 3 min at 93°C, 2 min at 72°C, and overnight at 37°C. Afterward, slides were washed once at 37°C with a 1:1 ratio of 2× SCCT: 50% formamide, once in 2× SCCT containing 10% Tween-20 and then three times in 2× SCCT. Slides were then blocked in 1% BSA, and fluorescently labeled anti-biotin and anti-digoxigenin antibodies were applied for 1 h at room temperature. After DAPI staining, the slides were mounted in Vectashield.

### RT-qPCR

RNA was isolated from 50 adult hermaphrodites (20 h after L4 stage or 20 h after irradiation with 120 Gy) per genotype using TRIzol (TriFast Peqlab; VWR), according to the manufacturer's protocol. cDNA was synthesized using the Invitrogen Super-Script III First-strand synthesis system with random hexamers according to the manufacturer's instructions. RT-qPCR was performed using the Bioline SensiFAST SYBR No-ROX Kit master mix with 20 ng cDNA in a final volume of 20 µl and three-step cycling.

The primer pair for *egl-1* was forward: 5′-CACCTTTGCCTC AACCTCTT-3′, reverse: 5′-GCTGATCTCAGAGTCATCAAAA-3′; and for *htp-1* (the reference gene for normalization) was forward: 5′-AACCATCTACGACGAATCGCT-3′, reverse: 5′-CATGCA TTTCAGCTTTTCAGTGATG-3′.

For all genotypes, three biological replicates corresponding to three independent RNA extractions and RT-qPCR reactions were performed.

### Gene editing using the CRISPR/Cas9 system
#### top-3 *mutant*

Two homology regions (the left and right homology regions) from the *top-3* locus, each comprising ~500 bp (the right homology region is located in exon 1 and the left homology region in exon 5), were amplified from genomic DNA, fused to the *unc-119* gene locus (from the pCFJ150 vector), and cloned using the Zero Blunt TOPO PCR Cloning Kit (Thermo Fisher Scientific) according to the manufacturer's instructions. The construct was sequence-verified and injected into *unc-119 (ed9)* worms. The three guides used for the injections were inserted into the pU6:: klp-12 single-guide RNA (sgRNA) vector (Addgene; 46170), as previously described (Friedland et al., 2013).

sgRNAs used for *top-3* knockout were as follows: sgRNA1, 5′-GAGAAGAATGACGTGGCAAAGG-3′; sgRNA2, 5′-GTCGCTGCG ATTTTATCAAATGG-3′; and sgRNA3, 5′-AATGAAAAGAGCCTT ATTTGTGG-3′. This resulted in gene disruption after amino acid 33. A preexisting mutation in the protospacer adjacent motif sequence resulted in a change toof amino acid 7. In addition, the unc-119 insertion led to the deletion of 171 amino acids (from amino acid 34 to 205, corresponding to the first four exons).

#### gfp::cku-80, rad-51::degron *and* top-3::ha::degron

*gfp::cku-80*, *rad-51::degron* and *top-3::ha::degron* were generated using a published protocol (Paix et al., 2015). To generate the *rad-51::degron* strain, the repair template was composed of two ssDNA molecules (163 bp and 128 bp) with a 35-bp overlap between them and a 35-bp overlap with homology to the *rad-51*

sequence. Into this sequence, we inserted a linker (5′-GGTGGC AGTGGAGGTGGCAGTGGA-3′) and the degron sequence (5′-ATGCCTAAAGATCCAGCCAAACCTCCGGCCAAGGCACAAGTT GTGGGATGGCCACCGGTGAGATCATACCGGAAGAACGTGATG GTTTCCTGCCAAAAATCAAGCGGTGGCCCGGAGGCGGCGGCG TTCGTGAAG-3′) directly before the stop codon.

The same strategy was used to generate the *top-3::ha::degron*. The injection mix was done as previously described (Paix et al., 2015). CRISPR events were followed by PCR analysis of the progeny, with candidate strains verified by sequencing.

For the *gfp::cku-80* repair template, a 30-bp homologous sequence from the N-terminus was fused to GFP to generate an N-terminally tagged CKU-80 protein with a linker sequence (5′-GGTGGCAGTGGA-3′) between the GFP and CKU-80 ORFs. The following CRISPR RNA (crRNA) was used: for *gfp::cku-80*, 5′-TACAGGAATGCCGCCTAAAAAGG-3′.

To generate the *rad-51::degron* strain, the following guide RNA was used: crRNA for the *rad-51::degron*, 5′-TGGTATTGAGGACGC ACGCGAGG-3′.

To generate the *top-3::ha::degron* strain, the repair template was composed of two ssDNA molecules (135 bp and 170 bp) with a 35-bp overlap between them and a 35-bp overlap with homology to the *top-3* sequence. The TIR-1 binding peptide was inserted in an internal position (between Gly 635 and Gly 636), together with an HA tag (5′-TACCCATACGATGTTCCAGATTAC GCT-3′). Three linkers were inserted in the following order: linker (5′-AGTAGCGGAAGT-3′)–HA–linker (5′-GGTGGCAGTG-GA-3′)–degron–linker (5′-AGTAGCGGAAGT-3′). The following crRNA was used for the *top-3::ha::degron*: 5′-CCTGGAGGTGGT GGTGGGGGAGG-3′.

CRISPR events were followed by PCR analysis of the progeny, and candidate strains were verified by sequencing.

### Auxin treatment

A 440-mM auxin stock solution was prepared in pure ethanol and stored at 4°C. *rad-51::degron* worms were transferred to seeded NGM plates, with both the plates and bacterial (*E. coli* strain OP50) suspension containing 1 mM auxin. At the indicated time points, samples were taken for analysis. *rad-51::degron* display a complete depletion of RAD-51 already after 30 min of auxin exposure (Fig. S3 A).

*top-3::ha::degron* young adult worms were transferred to auxin plates for 10 h before irradiation (120 Gy); apoptosis was quantified 20 h later (control untreated/unirradiated worms were subjected to the same timings). *top-3::ha::degron* display a complete depletion of HA foci after 2 h of auxin exposure (Fig. S3 C).

For all auxin treatments, NGM plates suplemented with 0.25% ethanol was used as the control.

### EdU treatment

At 20 h after L4 stage, worms were incubated in M9 buffer supplemented with 20 μM EdU for 15 min. Worms were then transferred to seeded plates, and the gonads were dissected at the indicated time points after EdU incorporation. Prior to DAPI staining, the Click-iT reaction was performed according to the manufacturer's instructions.

### Statistical analysis

Statistical analysis used the Mann–Whitney test in Prism 6 software (GraphPad) and Microsoft Excel. The number of events was recorded, and error bars (mean ± SD) and statistically significant differences were determined.

### Online supplemental material

Fig. S1, which is related to Fig. 1, shows that *top-3* mutants display joint DNA structures at diakinesis that are independent of the major and alternative DNA repair pathways. Fig. S2, related to Figs. 1, 2, and 3, shows that *top-3(jf101)* DNA lesions do not trigger a proper apoptotic response. Fig. S3, which is related to Figs. 4 and 5, shows that in *top-3*, meiotic progression is not delayed. Fig. S4, related to Fig. 5, shows that the GFP::CKU-80 transgene is functional. Table S1 lists reagents and resources.

## Acknowledgments

We are indebted to Angela Graf for technical support. We thank Antoine Baudrimont for drawing the model; Maria Velkova for experimental assistance; and Anton Gartner, Nicola Silva, Dimitra Paouneskou, and Maria Velkova for helpful suggestions. We are grateful to the Gartner, Zetka, Conradt (University College London, London, UK), and Hyman laboratories for reagents.

The V. Jantsch laboratory is funded by the Austrian Science Fund (P 31275-B28 and W1238). P. Barraud acknowledges financial support from the Centre National de la Recherche Scientifique. Some strains were provided by the *Caenorhabditis* Genetics Center, which is funded by the National Institutes of Health Office of Research Infrastructure Programs (P40OD010440).

The authors declare no competing financial interests.

Author contributions: M.R. Dello Stritto and V. Jantsch conceived all experiments and wrote the paper. M.R. Dello Stritto conducted all experiments, except for a few strains that were constructed by B. Bauer. B Bauer did the CED-1::GFP quantification (± irradiation). P. Barraud designed the *top-3::ha::degron* line.

Submitted: 9 December 2020

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

# Supplemental material

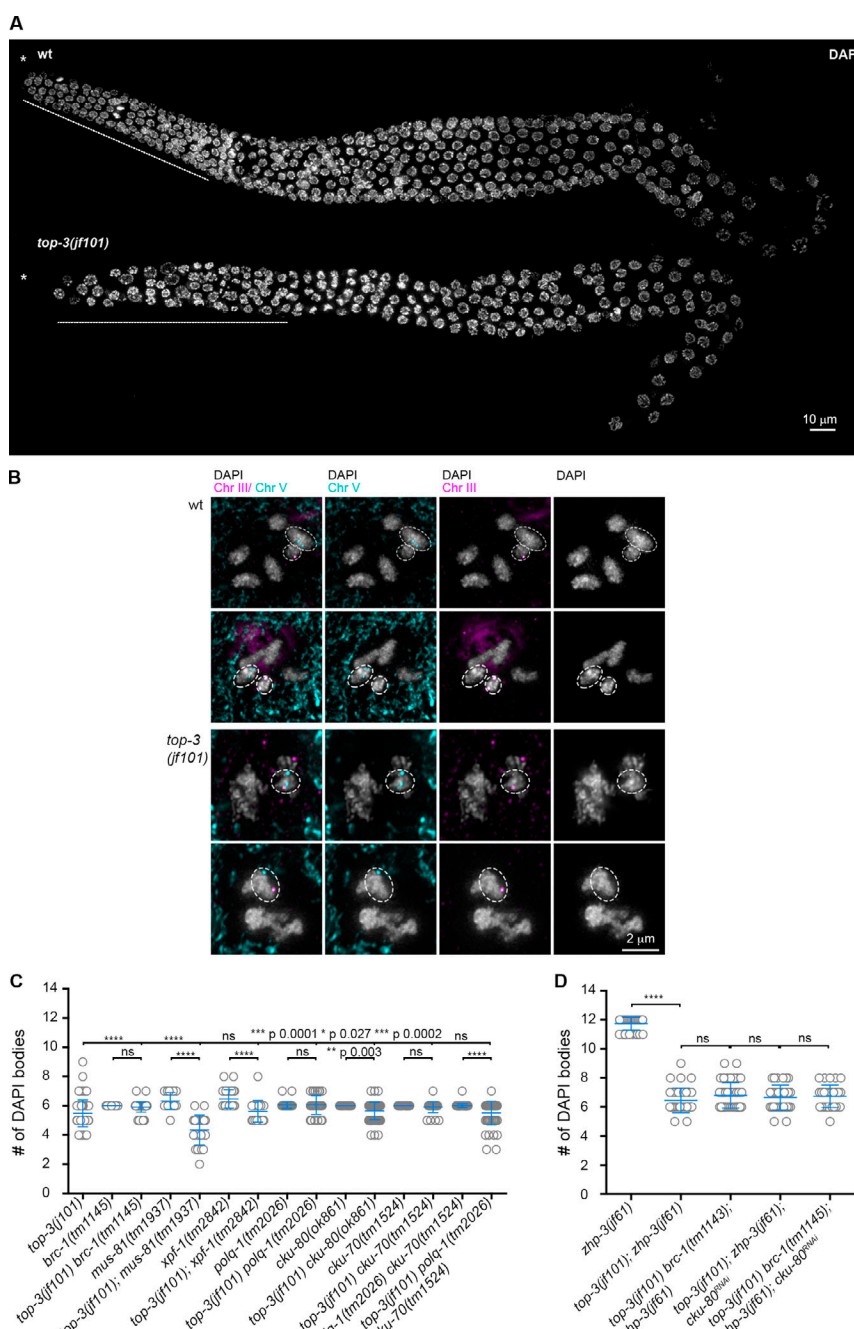

Figure S1. ***top-3* mutants display joint DNA structures at diakinesis that are independent of the major and alternative DNA repair pathways.** Related to Fig. 1. **(A)** Representative images of gonads stained with DAPI of the indicated genotypes. *, Mitotic tip. Dashed line marks the mitotic gonad region. **(B)** FISH analysis of wt and *top-3(jf101)* diakinesis using two different probes: chromosome III (magenta) and chromosome V (cyan). Dashed circles highlight single DAPI bodies that were assessed in the image stacks before projection. **(C and D)** Quantification of DAPI-stained bodies in −1 diakinesis oocyte nuclei of different genotypes. Scatter plots indicate the mean ± SD; n = number of oocytes for each genotype. **(C)** *top-3(jf101)* (as shown in Fig. 1 C), 5.5 ± 0.9, n = 66; *brc-1(tm1145)*, 6 ± 0, n = 20; *brc-1(tm1145) top-3(jf101)*, 5.9 ± 0.3, n = 104; *mus-81(tm1937)*, 6.3 ± 0.6, n = 21; *mus-81(tm1937); top-3(jf101)*, 4.3 ± 1.02, n = 26; *xpf-1(tm2842)*, 6.5 ± 0.6, n = 30; *xpf-1(tm2842); top-3(jf101)*, 5.6 ± 0.7, n = 21; *polq-1(tm2026)*, 6.06 ± 0.2, n = 35; *polq-1(tm2026) top-3(jf101)*, 6.06 ± 0.6, n = 34; *cku-80(ok861)*, 6 ± 0, n = 26; *top-3(jf101) cku-80(ok861)*, 5.6 ± 0.6, n = 88; *cku-70(tm1524)*, 6 ± 0, n = 38; *top-3(jf101) cku-70(tm1524)*, 5.9 ± 0.4, n = 34; *polq-1(tm2026) cku-70(tm1524)*, 6.02 ± 0.1, n = 65; and *top-3(jf101) polq-1(tm2026) cku-70(tm1524)* 5.5 ± 0.8, n = 55. **(D)** *zhp-3(jf61)*, 11.7 ± 0.4, n = 38; *zhp-3(jf61); top-3(jf101)*, 6.5 ± 0.8, n = 35; *zhp-3(jf61); brc-1(tm1145) top-3(jf101)*, 6.8 ± 0.9, n = 45; *zhp-3(jf61); top-3(jf101); cku-80^{RNAi}*, 6.7 ± 0.9, n = 32; and *zhp-3(jf61); top-3(jf101) brc-1(tm1145); cku-80^{RNAi}*, 6.7 ± 0.8, n = 31. Note that the mean number of DAPI diakinesis bodies is similar for all indicated double and triple mutants and that the configuration of DAPI bodies in diakinesis (undefined DNA masses) resembles those seen in *top-3*, also in the absence of CO formation. Error bars indicate the mean ± SD per genotype. Comparison calculated using the Mann–Whitney test: *top-3(jf101)* versus *brc-1(tm1145) top-3(jf101)*, ****, P < 0.0001; *top-3(jf101)* versus *mus-81(tm1937); top-3(jf101)*, ****, P < 0.0001; *top-3(jf101)* versus *xpf-1(tm2842); top-3(jf101)*, ns; *top-3(jf101)* versus *polq-1(tm2026) top-3(jf101)*, ***, P = 0.0001; *top-3(jf101)* versus *top-3(jf101) cku-80(ok861)*, *, P = 0.027; *top-3(jf101)* versus *top-3(jf101) cku-70(tm1524)*, ***, P = 0.0002; *top-3(jf101)* versus *top-3(jf101) polq-1(tm2026) cku-70(tm1524)*, ns. Chr, chromosome; wt, wild-type.

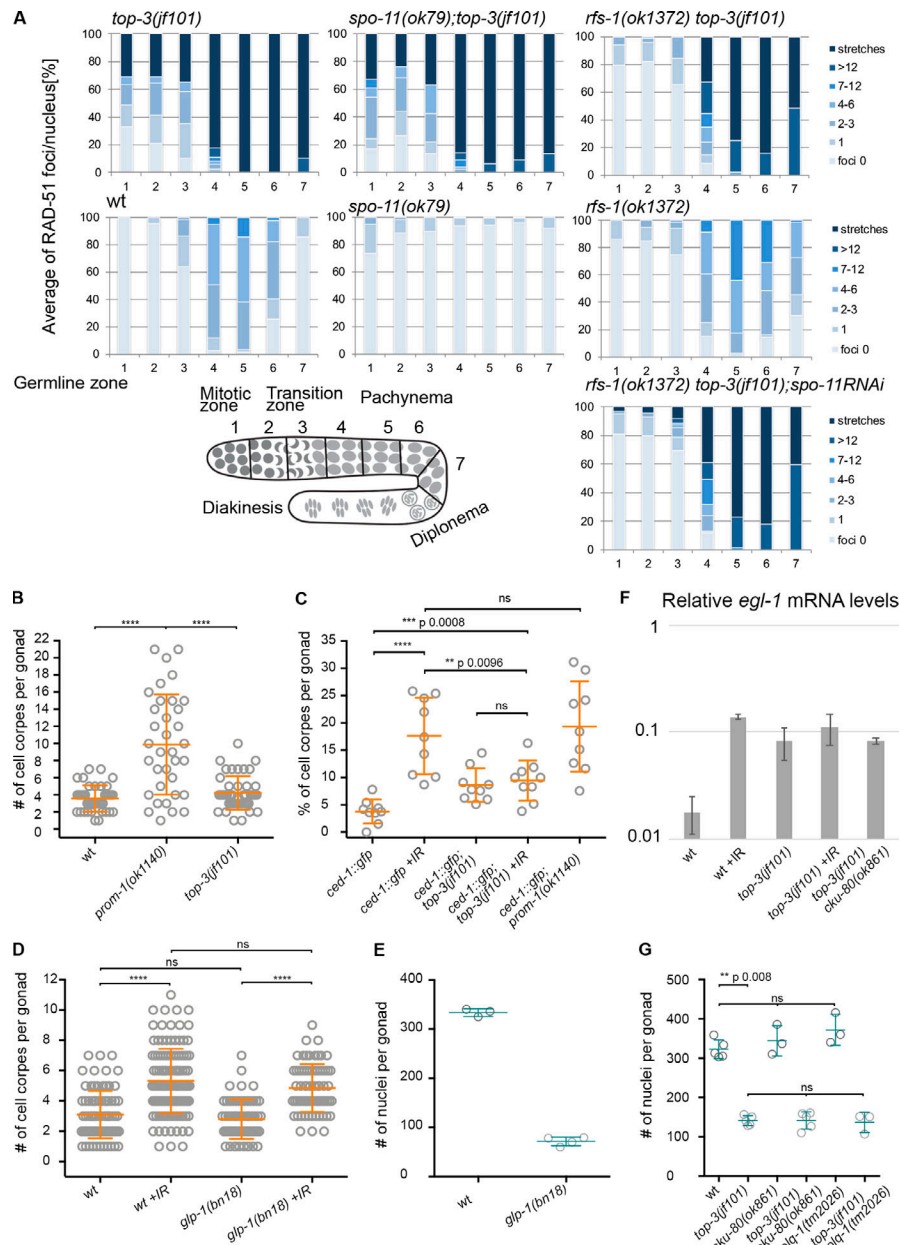

Figure S2. **top-3(jf101) DNA lesions do not trigger a proper apoptotic response.** Related to Figs. 1, 2, and 3. **(A)** The percentage of RAD-51 foci per nucleus in each of the seven zones in the indicated genotypes. For each zone, the average number of RAD-51 foci/nucleus was calculated from three gonads/genotype. Bottom left: Schematic representation of the *C. elegans* gonad divided into seven equal zones. **(B)** Apoptosis quantification with SYTO-12. Apoptotic levels in the *top-3* mutant were compared with those in the *prom-1* mutant, which has elevated DNA damage–induced apoptosis. wt and *top-3* have similar levels of apoptosis (shown in Fig. 2 D), but levels are significantly different in *prom-1* (9.9 ± 5.9, n = 35 scored gonads) and *top-3(jf101)*. ****, P < 0.0001, calculated using the Mann–Whitney test. **(C)** Scatter plot displaying the percentage of apoptotic nuclei assessed by CED-1::GFP staining, normalized to the total number of nuclei in the last 10 cell rows of the gonad in the indicated genotypes. Nine gonads were analyzed per genotype. Error bars indicate the mean ± SD per genotype: *ced-1::gfp*, 3.8 ± 2.2; *ced-1::gfp* +IR, 17.6 ± 7; *ced-1::gfp; top-3(jf101)*, 8.6 ± 3; *ced-1::gfp; top-3(jf101)* +IR, 9.4 ± 3.7; and *ced-1::gfp; prom-1(ok1140)*, 19.3 ± 8.3. ****, P < 0.0001, calculated using the Mann–Whitney test. **(D)** Apoptosis quantification with SYTO-12 of *glp-1* mutants with and without irradiation compared with wt. *glp-1* is a temperature-sensitive allele: the worms are grown at 16°C and shifted to 25°C at the L4 stage. Experiments on *glp-1* were performed at 25°C and 7 h after irradiation. Scatter plots indicate the mean ± SD; number of apoptotic nuclei for each genotype: wt, 3.1 ± 1.6, n = 82; wt +IR, 5.3 ± 2.1, n = 122; *glp-1(bn18)*, 2.8 ± 1.3, n = 63; and *glp-1(bn18)* +IR, 4.9 ± 1.6, n = 70. n = number of scored gonads. ****, P < 0.0001, calculated using the Mann–Whitney test. **(E)** Quantification of the total number of germline nuclei from the mitotic region to late pachynema. Scatter plots indicate the mean ± SD. Three gonads per genotype were analyzed: wt, 333.7 ± 7.8; and *glp-1(bn18)*, 71.75 ± 8.8. **(F)** Relative *egl-1* mRNA levels, as quantified by RT-qPCR, in wt, *top-3(jf101)*, and *top-3(jf101) cku-80(ok861)* after irradiation and unirradiated. Numbers correspond to the average of at least three independent RNA extractions and RT-qPCR reactions. Error bars indicate the SD. **(G)** Quantification of the total number of germline nuclei from the mitotic region to late pachynema. Scatter plots indicate the mean ± SD. Three to five gonads were quantified for each genotype. wt and *top-3(jf101)* are identical to Fig. 1 D; wt, 322.8 ± 23.7; *top-3(jf101)*, 141.4 ± 12.5; *cku-80(ok861)*, 344.3 ± 38.5; *top-3(jf101) cku-80(ok861)*, 141.6 ± 22.2; *polq-1(tm2026)*, 372 ± 39.4; and *top-3(jf101) polq-1(tm2026)*, 136.7 ± 25.7. P values calculated using the Mann–Whitney test. wt, wild-type.

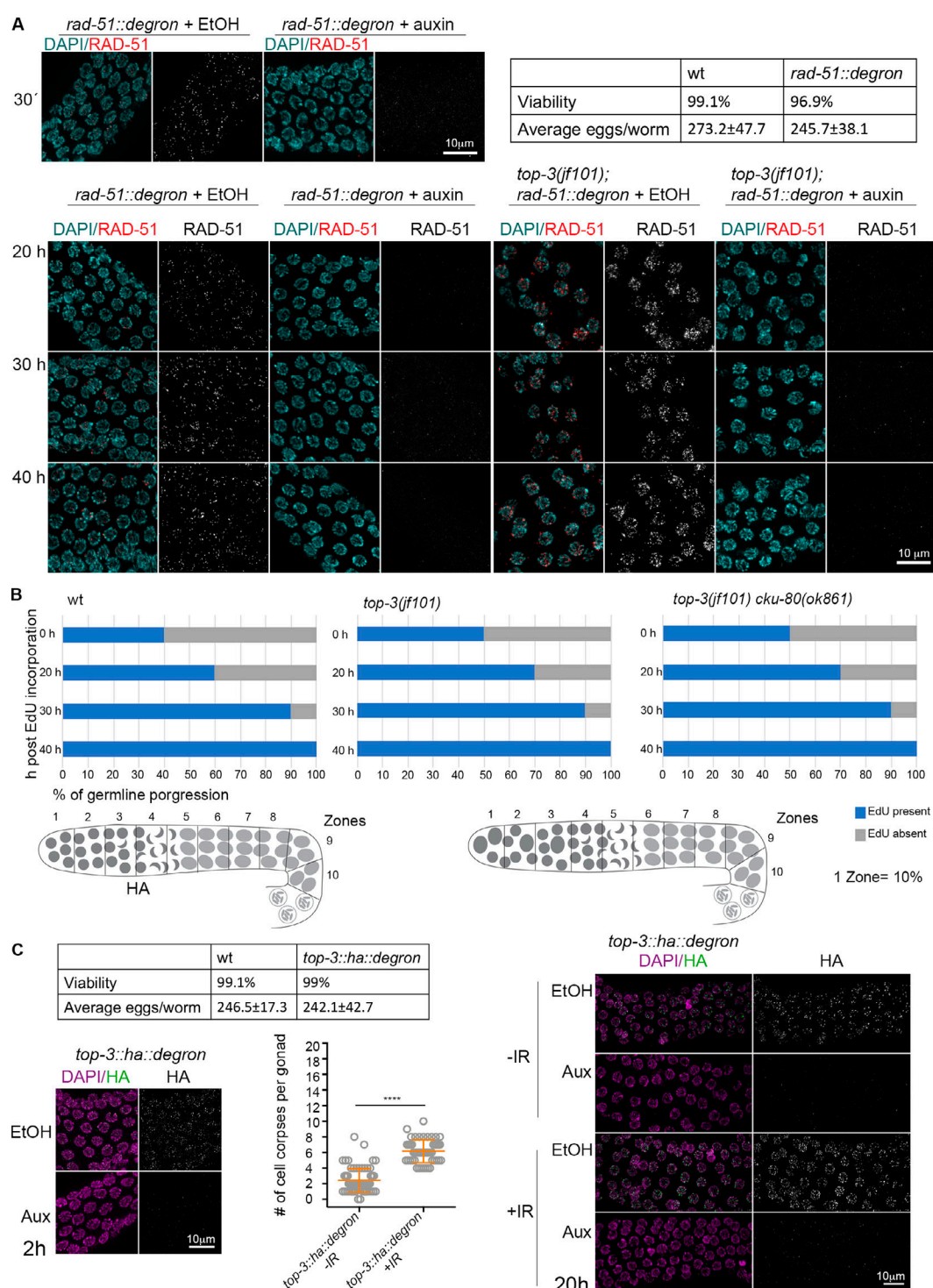

Figure S3. **In *top-3*, meiotic progression is not delayed.** Related to Fig. 4 and Fig. 5. **(A)** Top: representative *rad-51::degron* nuclei in pachynema stained with DAPI (cyan) and anti-RAD-51 (red) exposed to ethanol (EtOH) or auxin for 30 min, time needed for RAD-51 depletion. Hatch rates and brood sizes of *rad-51:: degron* (n = 15 worms) and wt (n = 12 worms). Bottom: representative *rad-51::degron* and *top-3(jf101); rad-51::degron* nuclei in pachynema stained with DAPI (cyan) and anti-RAD-51 (red) exposed to ethanol (EtOH) or auxin at different time points. **(B)** Top: The charts indicate EdU incorporation in the germline, divided into 10 equal zones. EdU incorporation was quantified: EdU present (blue) or EdU absent (gray). For the indicated genotypes, four time points were analyzed. Bottom: Schematic representation of wt and mutant gonads divided into 10 equal zones to show the prolonged mitotic zone in *top-3*. **(C)** Left: Functionality of the *top-3::ha::degron* strain. Hatch rates and brood sizes of *top-3::ha::degron* (n = 23 worms); and wt (n = 10 worms). Representative *top-3::ha:: degron* nuclei in pachynema, stained with DAPI (magenta) and anti-HA (green) exposed to EtOH or auxin for 2 h, time needed for TOP-3 depletion. Apoptosis quantification of *top-3::ha::degron* with and without irradiation with SYTO-12. Scatter plots indicate the mean ± SD. n = number of gonads scored: *top-3::ha:: degron* not irradiated, 2.4 ± 1.5, n = 72; *top-3::ha::degron* after irradiation, 6.2 ± 1.5, n = 48. ****, P < 0.0001, calculated using the Mann–Whitney test. Right: Representative *top-3::ha::degron* nuclei in pachynema, stained with DAPI (magenta) and anti-HA (green) exposed to EtOH or auxin with and without irradiation (120 Gy). Aux, auxin; wt, wild-type.

Figure S4. **The GFP::CKU-80 transgene is functional.** Related to Fig. 5. **(A)** Top: Survival after irradiation was assessed by determining larval arrest, the number of unhatched eggs, and viability (i.e., development to adulthood): *cku-80*, n = 323 embryos; *gfp::cku-80*, n = 333 embryos; and wt, n = 296 embryos. Note that the tagged version of CKU-80 appears to be functional because the mutant shows little larval arrest, and most progeny are viable. Bottom: Quantification of the fluorescence signal during different prophase I stages in wt (control) and *him-6(ok412)*. The scatter plot shows the mean ± SD. Mean values for the different stages: wt, 132.0–144.7–122.4; *him-6(ok412)*, 1,275–1,224–4,795. **(B)** Apoptosis quantification in *top-3::ha::degron* and *cku-80(ok861) top-3::ha::degron* in the absence (orange) or presence (green) of auxin using SYTO-12 with and without irradiation (120 Gy). Scatter plots indicate the mean ± SD. n = number of gonads scored: *top-3::ha::degron* (not irradiated) ethanol, 2.5 ± 1.3, n = 67; auxin, 2.2 ± 0.9, n = 58; *top-3::ha::degron* (after irradiation) ethanol, 5 ± 1.9, n = 51; auxin, 3.6 ± 1.6, n = 49; *cku-80(ok861) top-3::ha::degron* (not irradiated) ethanol, 2.5 ± 1.2, n = 60; auxin, 2.2 ± 1.1, n = 52; *cku-80(ok861) top-3::ha::degron* (after irradiation) ethanol, 6 ± 2, n = 44; auxin, 7 ± 2.5, n = 42. **(C)** Representative *cku-80(ok861) top-3::ha::degron* and *him-6(ok412); top-3::ha::degron* nuclei in pachynema, stained with DAPI (magenta) and anti-HA (green) exposed to ethanol (EtOH) or auxin with and without irradiation (120 Gy). **(D)** Apoptosis quantification using SYTO-12 in the indicated genotypes. Scatter plots indicate the mean ± SD. n = number of gonads scored. wt, 3.6 ± 1.5, n = 39 (quantification as described for Fig. 2 D); *mus-81(tm1937)*, 3.2 ± 1.5, n = 47; *top-3(jf101) mus-81(tm1937)*, 3.7 ± 1.7, n = 42; *top-3(jf101)*, 4.2 ± 1.9, n = 43 (quantification as described for Fig. 2 D); *cku-80(ok861) top-3(jf101)*, 8.4 ± 3, n = 47 (quantification as described for Fig. 6 A). **(E)** Quantification of DAPI-stained bodies in –1 diakinesis oocytes of different genotypes treated with *spo-11^RNAi*. Error bars indicate the mean ± SD. n = number of diakinesis nuclei scored: *top-3(jf101); spo-11^RNAi*, 10.8 ± 1.4, n = 21; *top-3(jf101) cku-80 (ok861); spo-11^RNAi*, 10.35 ± 1.4, n = 23; and *top-3(jf101) rfs-1(ok1373); spo-11^RNAi*, 11.1 ± 1.3, n = 21. Aux, auxin; wt, wild-type.

Table S1, which is provided online, lists reagents and resources.

