## [Peer Review File · The Journal of Cell Biology]

DNA Topoisomerase 3 is required for efficient germ cell quality control in *Caenorhabditis elegans*

Maria Rosaria Dello Stritto, Bernd Bauer, Pierre Barraud, and Verena Jantsch

Corresponding Author(s): Verena Jantsch, Max Perutz Labs; University of Vienna; Vienna Biocenter

Review Timeline:

Submission Date:	2020-12-09
Editorial Decision:	2021-01-12
Revision Received:	2021-02-21
Editorial Decision:	2021-03-09
Revision Received:	2021-03-10

Monitoring Editor: Arshad Desai

Scientific Editor: Melina Casadio

Transaction Report:

DOI: <https://doi.org/10.1083/jcb.202012057>

January 12, 2021

Re: JCB manuscript #202012057

Prof. Verena Jantsch
Max Perutz Labs; University of Vienna; Vienna Biocenter
Dr. Bohrgasse 9
Vienna 1030
Austria

Dear Verena,

Thank you for submitting your manuscript entitled "DNA Topoisomerase 3 is required for efficient germ cell quality control in *Caenorhabditis elegans*". The manuscript has been evaluated by expert reviewers, whose reports are appended below. Unfortunately, after an assessment of the reviewer feedback, our editorial decision is against publication in JCB.

The reviewers and we found the inability of top-3 mutants to elevate apoptosis interesting; however, the referees also asked how this occurs mechanistically. Reviewers #1 and #2 provide direct and excellent suggestions, pursuit of which would help elevate the impact of the work. In addition, the reviewers shared experimental concerns and points for clarification. Of particular importance is Reviewer #1's comment on the need for a clearer Introduction that focuses on the major question addressed (how genomic lesions trigger apoptosis) as well as a Results section that is written for a broad cell biology audience that is largely unfamiliar with *C. elegans* genetics and nomenclature.

The feedback from the reviewers echoed our own sentiments at the time of initial editorial evaluation and indicate that the work needs extension to be suitable for the broad audience of the JCB: the observation that top-3 mutations block apoptosis needs further mechanistic investigation. In addition, one of us (A.D.) felt it important to consider whether the rate of germline nuclei flux (i.e., the rate at which nuclei progress through the germline) is differentially affected in top-3 mutants versus the double mutant states that restore apoptosis. A revision that addresses possibilities raised by the reviewers to develop the major new finding on control of apoptosis may be appropriate for reconsideration at JCB. If you are interested in resubmitting to JCB, we would be happy to weigh in on a revision strategy or update - possibly with reviewer input - through the appeal workflow. This may be helpful to ensure you do not embark on time- and resource-consuming revisions that may not be sufficient for a successful resubmission to the journal. You may contact the journal office or submit an appeal directly through our manuscript submission system. Please note that priority and novelty would be reassessed at resubmission.

However, we also acknowledge that such an effort would require significant new experimental analysis and may not yield clear answers, despite constraining possible models. We understand that it is up to you and your colleagues to decide how to develop the work. If you wish to expedite publication of the current data, it may be best to pursue publication at another journal. Our office can transfer the reviews to any other journal upon request.

Regardless of how you choose to proceed, we hope that the comments below will prove

constructive as your work progresses. You can contact the journal office with any questions, cellbio@rockefeller.edu or call (212) 327-8588. Thank you for thinking of JCB as an appropriate place to publish your work.

Sincerely,

Arshad Desai, PhD
Editor, Journal of Cell Biology

Melina Casadio, PhD
Senior Scientific Editor, Journal of Cell Biology

Reviewer #1 (Comments to the Authors (Required)):

In this manuscript entitled "DNA Topoisomerase 3 is required for efficient germ cell quality control in *Caenorhabditis elegans*", Stritto et al. describe a novel role of DNA Topoisomerase 3, TOP-3, in germ cell quality control. TOP-3 is a component of the STR/BTR complex, which functions to migrate and decatenate double Holliday Junctions. Here the authors explored the role of TOP-3 in the decision to induce apoptosis in response to persistent DNA damage. They found that mutant worms lacking TOP-3 accumulate DNA lesions in both pre-meiotic and meiotic regions of the germline. However, top-3 mutants are unable to induce apoptosis, although the initial CEP-1 (p53)-dependent pathway is activated. This phenotype is unique to the top-3 mutant, as worm strains lacking other members of the BTR complex, such as him-6 and rmh-1/2, can induce apoptosis in response to DNA damage. Using time course experiments following auxin-mediated depletion of RAD-51, the authors showed that persistent RAD-51 in the pre-meiotic nuclei is responsible for preventing apoptosis in the top-3 mutant. The authors also showed that non-homologous end joining (NHEJ) or alternative NHEJ factors are upregulated in the top-3 mutant and that the downregulation of these factors enables the top-3 mutant to trigger apoptosis in a CEP-1-dependent manner. Thus, meiotic DNA breaks are repaired via the NHEJ pathways in top-3 mutants, and this contributes to the inefficient apoptotic response.

Although the data support the main conclusion of the paper, the writing of this manuscript makes it very difficult to recognize the significance of this work. No scientific question was raised in the Introduction, and it fails to highlight the knowledge gap that the authors are trying to address from the get-go. Furthermore, no mechanical insight is presented in Results and Discussion regarding how TOP-3 might function to evade DNA damage-induced apoptosis, independently of its role within the BTR complex. DNA lesions in top-3 mutants clearly activate the canonical DNA damage response, which results in CEP-1 (p53)-dependent expression of EGL-1 (BH3 only proteins). Is TOP-3 required for the pro-apoptotic pathway downstream of EGL-1, leading to caspase activation (e.g. what happens to cytochrome c release in top-3 mutants)? What about the ubiquitin signaling required for the DNA damage-induced apoptotic response? Have the authors examined UFD-2 foci in top-3 mutants?

Another major criticism is regarding the writing of the Results. The authors often jump to the conclusions without even describing experimental results (e.g. Figure 4C, lines 278-279). Please fully describe the experiments/results and explain why such conclusions can be deduced whenever appropriate. Also, many alleles have been used in complex genetic experiments without proper introductions. For a broad audience who may not be familiar with *C. elegans* gene names, please

introduce the genes used in the experiments (e.g. zhp-3 as a putative SUMO E3 ligase required for crossover formation; mus-81 as a structure-specific endonuclease; prom-1 as an F-box protein for SCF E3 ligase; glp-1 as a Notch receptor essential for the proliferation of germline stem cells, etc.).

Here are other comments:

- The first sentence in Short Summary has the word "germline" twice (line 24).
- Please consider revising the sentence in lines 37-40 in Abstract. It is too long and confusing.
- On page 4, lines 90-91, dJH is resolved either as crossover or non-crossover depending on the directionality of DNA cleavage by the resolvase, and both of these are outcomes of recombination.
- The paragraph on UFD-2 ubiquitin ligase (lines 115-118) does not fit very well into the flow. It is also vague how UFD-2 regulates RAD-51 dissociation. Clearly state that UFD-2 "promotes" RAD-51 dissociation.
- Line 127, based on the information provided in the Introduction, it is not clear how the activation of NHEJ can prevent the efficient culling of unhealthy oocytes.
- Given the similar level of apoptosis in wild-type vs. top-3 mutants, how can you explain the dramatic decrease in the nuclei number/gonad and number of eggs/worm in the top-3 mutant?
- Figure 1D was mentioned before Figure 1C in the main text. Please consider switching the order of these two Figures.
- On page 11, line 264, PGL-3 should be changed to PGL-1.
- In Figure 5, can the authors describe TOP-3 localization? Does it localize to the recombination intermediates, similarly to HIM-6 and RMH-1?
- On page 12, lines 305, please indicate which promoter was used to express TIR1 to deplete TOP-3 in meiosis.
- On page 14, line 356, references are needed for the statement that "The STR/BTR complex is not reported to make a major contribution to the resection steps in meiosis".
- In Discussion, lines 372-375 and 379-381 are stating the same thing.

Reviewer #2 (Comments to the Authors (Required)):

This is an interesting study that reports the unexpected finding that topoisomerase 3 (TOP-3), a member of the *C. elegans* BTR complex, is required for the execution of elevated levels of germline apoptosis in response to endogenous or exogenous DNA damage. Using the full spectrum of genetic, cell and molecular tools available, the authors show through multiple independent experiments that both mitotic and meiotic lesions fail to activate the full apoptotic response in the absence of TOP-3. Interestingly, TOP-3's function in the apoptotic response is independent of the other members of the BTR complex, suggesting a novel role for TOP-3, or the specific lesions generated in its absence, in apoptosis induction. The attenuated apoptosis response in top-3 mutants can be suppressed by inactivating several different recombination pathways: RAD-51 (HR), RFS-1 (HR), CKU-80 (NHEJ) or POLQ-1 (alt-EJ), suggesting that these pathways process top-3 lesions to generate an intermediate that somehow blocks apoptosis. These results provide mechanistic insight into quality control mechanisms in the germ line, which has implications for understanding infertility and birth defects. The following should be addressed:

1. Figure S1C: the authors examine DAPI-staining bodies (a readout of properly connected homologs) in top-3 mutants combined with mutations in various recombination pathways. In the results (lines 159-165), the authors state that there is no (variable) effect of removing these pathways, but in several instances, there are statistical differences. Please reword the results to

reflect the data. One might predict that since many of the mutations will elevate apoptosis in the top-3 background as reported in this manuscript, fewer diakinesis nuclei will have abnormal numbers of DAPI-staining bodies - I think that is the case in at least some of the double mutants, but the authors could help the reader both in the results section but also in the figure to highlight those differences.

2. The suppression of the top-3 apoptotic defect by multiple recombination pathways leads to a model whereby some intermediate generated by the absence of TOP-3 and acted on by these pathways abrogates apoptosis. Does removal of SSA (*xpf-1*) also suppress apoptosis? Have the authors examined RPA or RAD-51 (where appropriate) levels in the suppressed strains? This could provide insight into what is leading to the suppression of apoptosis in top-3 mutants.

3. The authors show elevated levels of CKU-80 in the top-3 mutant. Is this at the transcriptional or post-transcriptional level? Is POLQ-1 (or members of other pathways) also upregulated? Can the authors determine whether there are wide-scale transcriptional changes in the top-3 mutant that may explain the reduction in apoptosis?

4. Given that *egl-1* is upregulated, and SIR-2.1 and PGL-1 all behave similarly to wild type, I would like to see more discussion on how the authors envision a DNA intermediate leads to dampening of the apoptotic response.

Minor:

In summary, please remove the first germline (line 24)

On figure 2E, please add SYTO12 to top as was done for 2C and D.

In materials and methods, the quantification of RAD-51 refers to figure S1, but it should be S2 (line 445).

Reviewer #3 (Comments to the Authors (Required)):

1. This paper provides strong evidence that the *C. elegans* topoisomerase 3 (*top-3*) is necessary to prevent the generation of aberrant and lethal DNA lesions. Unlike other kinds of unrepaired DNA damage that arise in several known meiotic mutants, the lesions that are engendered in the *top-3* mutant are shown to be incapable of triggering increased germline apoptosis. While a subset of the apoptotic machinery does activate in *top-3* mutants, the authors suggest that the shunting of *top-3* lesions to the NHEJ and/or Alt-NHEJ pathways allows them to be repaired in a way that evades cell death. While the exact nature of the DNA lesions induced by *top-3* deletion is not clear, and the mechanism of cell death evasion is not shown, this paper sheds light on the importance of topoisomerase 3 in keeping chromatin intact, thereby enabling the normal pathways of germline quality surveillance to operate.

2. All the data are strongly supportive, save for the following: In Figure 4, the authors conclude, based on the timing of AID-induced degradation, that DNA damage capable of inducing apoptosis upon removal of RAD-51 is generated premeiotically in *top-3* mutants. The timing is referred to as "hours after RAD-51 degradation", but what we are shown is the time after auxin is administered. It should be stated or shown how quickly RAD-51 is degraded upon auxin treatment, to get a sense of the true time in which nuclei go without RAD-51 protein; similar considerations apply to the *top-3* AID experiments.

3. Minor comments:

The finding that *top-3* mutants start to activate the apoptosis program, but do not follow through

with it, is interesting and relevant to the paper's conclusions, but it is not discussed much. DNA lesions induced by the lack of top-3 are being recognized as damage and acted upon, and at least some of the lesions are then presumably repaired by NHEJ, but what stops apoptosis from continuing its normal course? If all damage is repaired before the "death zone", then why is e.g. egl-1 constitutively upregulated in top-3 mutants? A discussion of this point, even speculative, would be welcome.

58-68: The description of apoptosis contains elements that have little to do with the manuscript content ("export of mitochondria along microtubule cables"); this could be better matched to the assays that were done.

84: HR is used here without "homologous recombination" being spelled out first.

138: One fact that should be clarified is whether jf101 is recessive for the inviability phenotype?

142/147: Figure callout for 1D comes before 1C.

168: Here the manuscript says "We detected the accumulation of ssDNA coated with..." but this is a presumption based on the increase of immunofluorescence foci; ssDNA was not detected directly.

180: clarify that "these lesions cannot serve as substrates" refers to lesions found in the top-3 mutant condition.

589: Please clarify in the methods whether Cy3-dUTP (as is written) or EdU was used.

Summary of the most important changes:

We thank all the reviewers for the time they invested to propose improvements for our manuscript.

This is a summary of the major points we discussed with the editor:

1)

..."In addition, one of us (A.D.) felt it important to consider whether the rate of germline nuclei flux (i.e., the rate at which nuclei progress through the germline) is differentially affected in top-3 mutants versus the double mutant states that restore apoptosis."

This experiment is shown in Figure S3 B. It might have escaped your attention, but here we already showed that the rate of germline nuclei flux is not increased in top-3 cku-80 double mutants.

Suggestion by the editor.

2)

We made the strain *top-3::degron; him-6; Tir-1*. With this we assessed whether apoptosis is increased in this double mutant in comparison to *top-3::degron; Tir-1* alone. This showed us that in the absence of the topoisomerase the helicase itself generates aberrant recombination intermediates that apparently prefer alternative repair pathways. We have included these data and discussed them. These experiments certainly extend our mechanistic insight.

3)

We quantified apoptosis in *xpf-1 top-3* double mutants. This showed that yet another alternative micro homology mediated DNA repair pathway (SSA) can step in to process aberrant DNA repair intermediates caused by the absence of TOP-3. Suggestion by Reviewer 2.

4)

We quantified RAD-51 in strains where the apoptosis block is relieved. See below. Suggestion by Reviewer 2.

5)

We have requested the UFD-2 antibody to assess whether ubiquitin signaling required for the DNA damage-induced apoptotic response is activated? We have tried to detect UFD-2 foci in *top-3* mutants, see below. We have indeed tried more assays to assess the extent of apoptosis induction (eg activated caspase assays), however some of these assays are too unreliable for precise quantification. Physiological apoptosis is taking place in *top-3*, therefore we do not expect a yes/no answer. Only very robust assays and markers can be employed (eg., PGL-1, used in the ms was one of these). Suggestion by Reviewer 1.

6)

We have included the timing of degradation in the AID experiments (*rad-51::degron* and *top-3::degron* and *top-3::degron cku-80*). Suggestion by Reviewer 3.

7)

As to the transcriptional changes in *top-3*: it is commonly believed that TOP3 beta is involved in removing obstacle on the DNA during transcription. The *C. elegans* genome also encodes a *top-3 beta* paralog (Y48C3A.14) and we believe that it is responsible to remove those obstacles. Comment to suggestion by Reviewer 2.

8)

We provide the TOP-3 foci quantification profile. Suggestion by Reviewer 1.

9)

We have carefully address all the suggestions concerning the text edits. Suggested by all reviewers.

10)

We have tried to perform qPCR on *cku-80* and *polq-1* and compared mRNA levels in wild type and *top-3*. We did not obtain very conclusive results and this would need a lot more analysis also on other recombination factors. We therefore prefer to not include this into the revision. Suggestion by Reviewer 2.

Reviewer #1 (Comments to the Authors (Required)):

In this manuscript entitled "DNA Topoisomerase 3 is required for efficient germ cell quality control in *Caenorhabditis elegans*", Stritto et al. describe a novel role of DNA Topoisomerase 3, TOP-3, in germ cell quality control. TOP-3 is a component of the STR/BTR complex, which functions to migrate and decatenate double Holliday Junctions. Here the authors explored the role of TOP-3 in the decision to induce apoptosis in response to persistent DNA damage. They found that mutant worms lacking TOP-3 accumulate DNA lesions in both pre-meiotic and meiotic regions of the germline. However, top-3 mutants are unable to induce apoptosis, although the initial CEP-1 (p53)-dependent pathway is activated. This phenotype is unique to the top-3 mutant, as worm strains lacking other members of the BTR complex, such as him-6 and rrmh-1/2, can induce apoptosis in response to DNA damage. Using time course experiments following auxin-mediated depletion of RAD-51, the authors showed that persistent RAD-51 in the pre-meiotic nuclei is responsible for preventing apoptosis in the top-3 mutant. The authors also showed that non-homologous end joining (NHEJ) or alternative NHEJ factors are upregulated in the top-3 mutant and that the downregulation of these factors enables the top-3 mutant to trigger apoptosis in a CEP-1-dependent manner. Thus, meiotic DNA breaks are repaired via the NHEJ pathways in top-3 mutants, and this contributes to the inefficient apoptotic response.

Although the data support the main conclusion of the paper, the writing of this manuscript makes it very difficult to recognize the significance of this work. No scientific question was raised in the Introduction, and it fails to highlight the knowledge gap that the authors are trying to address from the get-go. Furthermore, no mechanical insight is presented in Results and Discussion regarding how TOP-3 might function to evade DNA damage-induced apoptosis, independently of its role within the BTR complex. DNA lesions in top-3 mutants clearly activate the canonical DNA damage response, which results in CEP-1 (p53)-dependent expression of EGL-1 (BH3 only proteins). Is TOP-3 required for the pro-apoptotic pathway downstream of EGL-1, leading to caspase activation (e.g. what happens to cytochrome c release in top-3 mutants)? What about the ubiquitin signaling required for the DNA damage-induced apoptotic response? Have the authors examined UFD-2 foci in top-3 mutants?

Another major criticism is regarding the writing of the Results. The authors often jump to the conclusions without even describing experimental results (e.g. Figure 4C, lines 278-279). Please fully describe the experiments/results and explain why such conclusions can be deduced whenever appropriate. Also, many alleles have been used in complex genetic experiments without proper introductions. For a broad audience who may not be familiar with *C. elegans* gene names, please introduce the genes used in the experiments (e.g. zhp-3 as a putative SUMO E3 ligase required for crossover formation; mus-81 as a structure-specific endonuclease; prom-1 as an F-box protein for SCF E3 ligase; glp-1 as a Notch receptor essential for the proliferation of germline stem cells, etc.).

We have included the rationale for our study in the introduction, improved the results description and extended the discussion and highlighted that this work presents an as-yet undescribed requirement for topoisomerase 3 in mounting an effective apoptotic response, which has not been shown before. We followed your recommendation to better describe the C. elegans genes. As you can see in the overall summary, we have now more mechanistic insights why apoptosis is lower than expected in the top-3 mutant (unscheduled activity of HIM-6 might be causing the aberrant recombination intermediates that are rather processed by normally not used DNA repair pathways (see addition of analysis of top-3::degron; him-6 and top-3; xpf-1 and top-3; mus-81).

We have consulted with an expert for C. elegans apoptosis (Anton Gartner), who told us that in worms there is no evidence/assays for cytochrome c release.

We have requested the UFD-2 antibody. However, we did not obtain any staining patterns that would help us understand what is going on in top-3 mutants. See, below.

Here are other comments:

- The first sentence in Short Summary has the word "germline" twice (line 24).

done

- Please consider revising the sentence in lines 37-40 in Abstract. It is too long and confusing.

done

- On page 4, lines 90-91, dJH is resolved either as crossover or non-crossover depending on the directionality of DNA cleavage by the resolvase, and both of these are outcomes of recombination.

done

- The paragraph on UFD-2 ubiquitin ligase (lines 115-118) does not fit very well into the flow. It is also vague how UFD-2 regulates RAD-51 dissociation. Clearly state that UFD-2 "promotes" RAD-51 dissociation.

done

- Line 127, based on the information provided in the Introduction, it is not clear how the activation of NHEJ can prevent the efficient culling of unhealthy oocytes.

We have included this in the discussion.

- Given the similar level of apoptosis in wild-type vs. top-3 mutants, how can you explain the dramatic decrease in the nuclei number/gonad and number of eggs/worm in the top-3 mutant?

We assume there are fewer nuclei entering meiosis. The DNA damage signaling in the mitotic compartment seems to work and this slows down the cell cycle and produces fewer premeiotic nuclei. In Fig S3B we show that the flux of nuclei through meiosis is not really affected by the reduced number of nuclei.

- Figure 1D was mentioned before Figure 1C in the main text. Please consider switching the order of these two Figures.

done

- On page 11, line 264, PGL-3 should be changed to PGL-1.

done

- In Figure 5, can the authors describe TOP-3 localization? Does it localize to the recombination intermediates, similarly to HIM-6 and RMH-1?

We have added this.

- On page 12, lines 305, please indicate which promoter was used to express TIR1 to deplete TOP-3 in meiosis.

Done

- On page 14, line 356, references are needed for the statement that "The STR/BTR complex is not reported to make a major contribution to the resection steps in meiosis".

done

- In Discussion, lines 372-375 and 379-381 are stating the same thing.

done

Reviewer #2 (Comments to the Authors (Required)):

This is an interesting study that reports the unexpected finding that topoisomerase 3 (TOP-3), a member of the *C. elegans* BTR complex, is required for the execution of elevated levels of germline apoptosis in response to endogenous or exogenous DNA damage. Using the full spectrum of genetic, cell and molecular tools available, the authors show through multiple independent experiments that both mitotic and meiotic lesions fail to activate the full apoptotic response in the absence of TOP-3. Interestingly, TOP-3's function in the apoptotic response is independent of the other members of the BTR complex, suggesting a novel role for TOP-3, or the specific lesions generated in its absence, in apoptosis induction. The attenuated apoptosis response in top-3 mutants can be suppressed by inactivating several different recombination pathways: RAD-51 (HR), RFS-1 (HR), CKU-80 (NHEJ) or POLQ-1 (alt-EJ), suggesting that these pathways process top-3 lesions to generate an intermediate that somehow blocks apoptosis. These results provide mechanistic insight into quality control mechanisms in the germ line, which has implications for understanding infertility and birth defects. The following should be addressed:

1. Figure S1C: the authors examine DAPI-staining bodies (a readout of properly connected homologs) in top-3 mutants combined with mutations in various recombination pathways. In the results (lines 159-165), the authors state that there is no (variable) effect of removing these pathways, but in several instances, there are statistical differences. Please reword the results to reflect the data. One might predict that since many of the mutations will elevate apoptosis in the top-3 background as reported in this manuscript, fewer diakinesis nuclei will have abnormal numbers of DAPI-staining bodies - I think that is the case in at least some of the double mutants, but the authors could help the reader both in the results section but also in the figure to highlight those differences.

We have elaborated more on the zhp-3 top-3 double mutant. In this genotype one would clearly expect univalents, which is strikingly not the case. This mutant clearly shows that top-3 is needed to resolve a

lot of catenene structures (as reported in yeasts). Overall, the catenenes make the interpretation of the diakinesis structures hard. We have discussed this also in the discussion.

2. The suppression of the top-3 apoptotic defect by multiple recombination pathways leads to a model whereby some intermediate generated by the absence of TOP-3 and acted on by these pathways abrogates apoptosis. Does removal of SSA (xpf-1) also suppress apoptosis?

We did include this analysis. It would actually support the assumption that also SSA is a route of DNA repair in the top-3 mutant.

Have the authors examined RPA or RAD-51 (where appropriate) levels in the suppressed strains? This could provide insight into what is leading to the suppression of apoptosis in top-3 mutants.

We did. See examples below. We have not included this in the paper because the overall RAD-51 signal of the top-3 mutant (the signal imported through the replication problems) is too massive to allow for meaningful quantifications.

3. The authors show elevated levels of CKU-80 in the top-3 mutant. Is this at the transcriptional or post-transcriptional level? Is POLQ-1 (or members of other pathways) also upregulated? Can the authors determine whether there are wide-scale transcriptional changes in the top-3 mutant that may explain the reduction in apoptosis?

We gave the qPCR a try but this did not convey a very clear message. It would need a lot more analysis (also on HR repair factors), but given the limited resources and time, we believe this analysis is beyond the scope of this manuscript.

4. Given that egl-1 is upregulated, and SIR-2.1 and PGL-1 all behave similarly to wild type, I would like to see more discussion on how the authors envision a DNA intermediate leads to dampening of the apoptotic response.

We have included this in the discussion.

Minor:

In summary, please remove the first germline (line 24)

done

On figure 2E, please add SYTO12 to top as was done for 2C and D.

done

In materials and methods, the quantification of RAD-51 refers to figure S1, but it should be S2 (line 445).

done

Reviewer #3 (Comments to the Authors (Required)):

1. This paper provides strong evidence that the *C. elegans* topoisomerase 3 (top-3) is necessary to prevent the generation of aberrant and lethal DNA lesions. Unlike other kinds of unrepaired DNA damage that arise in several known meiotic mutants, the lesions that are engendered in the top-3 mutant are shown to be incapable of triggering increased germline apoptosis. While a subset of the apoptotic machinery does activate in top-3

mutants, the authors suggest that the shunting of top-3 lesions to the NHEJ and/or Alt-NHEJ pathways allows them to be repaired in a way that evades cell death. While the exact nature of the DNA lesions induced by top-3 deletion is not clear, and the mechanism of cell death evasion is not shown, this paper sheds light on the importance of topoisomerase 3 in keeping chromatin intact, thereby enabling the normal pathways of germline quality surveillance to operate.

2. All the data are strongly supportive, save for the following: In Figure 4, the authors conclude, based on the timing of AID-induced degradation, that DNA damage capable of inducing apoptosis upon removal of RAD-51 is generated premeiotically in top-3 mutants. The timing is referred to as "hours after RAD-51 degradation", but what we are shown is the time after auxin is administered. It should be stated or shown how quickly RAD-51 is degraded upon auxin treatment, to get a sense of the true time in which nuclei go without RAD-51 protein; similar considerations apply to the top-3 AID experiments.

In the revised manuscript we have added pictures to show the minimal time required for RAD-51 and TOP-3 degradation in the degraon lines.

3. Minor comments:

The finding that top-3 mutants start to activate the apoptosis program, but do not follow through with it, is interesting and relevant to the paper's conclusions, but it is not discussed much. DNA lesions induced by the lack of top-3 are being recognized as damage and acted upon, and at least some of the lesions are then presumably repaired by NHEJ, but what stops apoptosis from continuing its normal course? If all damage is repaired before the "death zone", then why is e.g. *egl-1* constitutively upgraded in top-3 mutants? A discussion of this point, even speculative, would be welcome.

We have included this in the discussion

58-68: The description of apoptosis contains elements that have little to do with the manuscript content ("export of mitochondria along microtubule cables"); this could be better matched to the assays that were done.

We have taken out the irrelevant sentences.

84: HR is used here without "homologous recombination" being spelled out first.

Indeed it is spelled out.

138: One fact that should be clarified is whether *jf101* is recessive for the inviability phenotype?

we keep the strain top-3(jf101)/hT2 in heterozygosity, therefore we do not think it is dominant. It is possible that some embryos die because of a zygotic defect.

142/147: Figure callout for 1D comes before 1C.

done

168: Here the manuscript says "We detected the accumulation of ssDNA coated with..." but this is a presumption based on the increase of immunofluorescence foci; ssDNA was not detected directly.

done

180: clarify that "these lesions cannot serve as substrates" refers to lesions found in the top-3 mutant condition.

done

589: Please clarify in the methods whether Cy3-dUTP (as is written) or EdU was used.

done

March 9, 2021

RE: JCB Manuscript #202012057R-A

Prof. Verena Jantsch
Max Perutz Labs; University of Vienna; Vienna Biocenter
Dr. Bohrgasse 9
Vienna 1030
Austria

Dear Prof. Jantsch,

Thank you for submitting your revised manuscript entitled "DNA Topoisomerase 3 is required for efficient germ cell quality control in *Caenorhabditis elegans*". You will see that the reviewers appreciated the revisions and now recommend publication. We would be happy to publish your paper in JCB pending final revisions necessary to meet our formatting guidelines (see details below) and pending edits to address the reviewers' final suggestions.

1) Titles, eTOC: Please consider the following revision suggestions aimed at increasing the accessibility of the work for a broad audience and non-experts.

Title: Germline activity of topoisomerase 3 is needed for apoptosis upon meiotic recombination failure

eTOC summary: A 40-word summary that describes the context and significance of the findings for a general readership should be included on the title page. The statement should be written in the present tense and refer to the work in the third person.

Suggested revisions to meet our preferred style (It should start with "First author name(s) et al..."):

Dello Stritto et al. provide evidence that DNA lesions in both germline mitotic and meiotic compartments are less capable of triggering apoptosis in the absence of topoisomerase 3. In topoisomerase 3 mutants, uncontrolled bloom helicase activity governs repair of defective recombination intermediates to evade apoptosis.

2) Statistical analysis: Error bars on graphic representations of numerical data must be clearly described in the figure legend. The number of independent data points (n) represented in a graph must be indicated in the legend. Statistical methods should be explained in full in the materials and methods. For figures presenting pooled data the statistical measure should be defined in the figure legends.

3) Materials and methods: Should be comprehensive and not simply reference a previous publication for details on how an experiment was performed. Please provide full descriptions in the text for readers who may not have access to referenced manuscripts.

- For all cell lines, vectors, constructs/cDNAs, Worm lines, etc. - all genetic material: please include

database / vendor ID (e.g., Addgene, ATCC, WormBase, etc.) or if unavailable, please briefly describe their basic genetic features *even if described in other published work or gifted to you by other investigators*

- Please include species and source for all antibodies, including secondary, as well as catalog numbers/vendor identifiers if available.
- Sequences should be provided for all oligos: primers, si/shRNA, gRNAs, etc.
- Microscope image acquisition: The following information must be provided about the acquisition and processing of images:
 - a. Make and model of microscope
 - b. Type, magnification, and numerical aperture of the objective lenses
 - c. Temperature
 - d. imaging medium
 - e. Fluorochromes
 - f. Camera make and model
 - g. Acquisition software
 - h. Any software used for image processing subsequent to data acquisition. Please include details and types of operations involved (e.g., type of deconvolution, 3D reconstitutions, surface or volume rendering, gamma adjustments, etc.).

4) Tables must be separated from the M&M or converted to paragraph form - if you wish to keep it as a table, the "Reagents and Resources Table" needs to be separate from the M&M and provided as an individual, editable file (e.g., Word, excel).

A. MANUSCRIPT ORGANIZATION AND FORMATTING:

Full guidelines are available on our Instructions for Authors page, <https://jcb.rupress.org/submission-guidelines#revised>. **Submission of a paper that does not conform to JCB guidelines will delay the acceptance of your manuscript.**

B. FINAL FILES:

-- High-resolution figure and video files: See our detailed guidelines for preparing your production-ready images, <https://jcb.rupress.org/fig-vid-guidelines>.

****The license to publish form must be signed before your manuscript can be sent to production. A link to the electronic license to publish form will be sent to the corresponding author only. Please take a moment to check your funder requirements before choosing the appropriate license.****

Thank you for this interesting contribution, we look forward to publishing your paper in Journal of Cell Biology.

Sincerely,

Arshad Desai, PhD
Editor, Journal of Cell Biology

Melina Casadio, PhD
Senior Scientific Editor, Journal of Cell Biology

Reviewer #1 (Comments to the Authors (Required)):

The authors have now addressed the previous concerns. I'm also pleased to see the additional data suggesting how the uncontrolled HIM-6 activity might be responsible for evading apoptotic response in top-3 mutants. Both Results and Discussion sections are much improved, and it's a lot easier to follow the logical flow of the paper.

Here are minor points on punctuation and statistical analysis:

1. Line 130, please change "Further analysis reveal" to "Further analysis reveals".
2. Line 151-152, for non-C. elegans audience, it will be helpful to label the mitotic region in Figure S1A.
3. Line 302, please insert a comma before "we generated".
4. Line 311, please insert a comma before "which correspond".
5. Line 333, please insert a space between the sentences, before "TOP-3 localizes".
6. Line 344, insert a comma after "when HR is compromised".
7. Lines 350-351, statistical analysis and p value are required to make a comparison for CKU-80 signal between wt and top-3 mutants. Please consider presenting the entire dataset in one graph and present the p values in Figure 5C.
8. Lines 355-356, this has to be supported by the statistical analysis comparing cku-80 vs. cku-80 top-3 in Figure 6A.
9. Line 423, "depletion of TOP-3 or RAD-51 in top-3 mutants" is a bit confusing as it reads like TOP-3 is depleted in top-3 mutants, which does not make sense.
10. Line 431, please insert a comma before "we have also performed".
11. Line 433, top-3 should be capitalized as the protein is degraded.
12. Line 447, fix the typo, zph-3 to zhp-3.
13. Line 460, please insert a space between the sentences, before "The him-6 helicase".

Reviewer #2 (Comments to the Authors (Required)):

The authors have done an excellent job addressing the previous reviews and the revised manuscript is significantly improved. Specifically, the authors extend their genetic, molecular and cell biological analyses to show that removal of the alternative SSA pathway (XPF-1) and HIM-6 (Bloom Helicase) also suppress the apoptotic phenotype of top-3 mutants (in addition to their previous analyses of NHEJ, alt-EJ, RAD-51, RFS-1). The authors have very carefully addressed flux and have normalized germline cell numbers to make a compelling case that in the absence of TOP-3, HIM-6 generates lesions that are repaired by alternative pathways thereby dampening the apoptotic response. The modifications to the text improve clarity for a general readership and addition of new experiments provides a more mechanistic understanding of the how TOP-3 dysfunction is unable to activate the apoptotic pathway. My comments are minor for additional clarity.

1. Line 131-132: "Germline depletion of HIM-6, CKU-70/80, POLQ-1 OR XPF-1" isn't really accurate - the only germline experiment was done with cku-80 and top-3::deg. Please reword to more accurately describe what was done.
2. Line 284: "Apoptosis block in top-3 is not only due to a failure to remove RAD-51". I recommend that this title be changed. Removal of RAD-51 restores apoptosis and the authors show this is due to removal of RAD-51 in the mitotic compartment, but not in meiosis. Perhaps specify that it is due to removal of RAD-51 in the mitotic compartment.
3. Line 289: "We could observe" → We observed
4. Line 308: Somewhere in here the authors should specifically refer to Supplemental Figure 3 showing efficient depletion of the rad-51::degron (similar to what was done for top-3) in addition to stating it in the materials and methods.
5. Line 371: "We wanted to test" → we tested
6. Line 379: Please qualify: "Thus, it is likely that the intermediates that . . ."
7. Line 414: "Based on our findings, we would like to propose a previously undescribed consequence of mutating topoisomerase 3 in the germline." You are not proposing this consequence, you show it and then propose a model for why TOP-3 is required for the full apoptotic response.
8. Line 453: "Would be that" → is that

In conclusion, the authors uncover a novel role for TOP-3 and HIM-6 with respect to the apoptotic response. These results provide insight into quality control mechanisms in the germ line, which has implications for understanding infertility and birth defects.